# Logarithmic Regret from Sublinear Hints

**Aditya Bhaskara**
Department of Computer Science
University of Utah
Salt Lake City, UT
bhaskaraaditya@gmail.com

**Ashok Cutkosky**
Dept. of Electrical and Computer Engineering
Boston University
Boston, MA
ashok@cutkosky.com

**Ravi Kumar**
Google Research
Mountain View, CA
ravi.k53@gmail.com

**Manish Purohit**
Google Research
Mountain View, CA
mpurohit@google.com

## Abstract

We consider the online linear optimization problem, where at every step the algorithm plays a point $x_t$ in the unit ball, and suffers loss $\langle c_t, x_t \rangle$ for some cost vector $c_t$ that is then revealed to the algorithm. Recent work showed that if an algorithm receives a *hint* $h_t$ that has non-trivial correlation with $c_t$ before it plays $x_t$, then it can achieve a regret guarantee of $O(\log T)$, improving on the bound of $\Theta(\sqrt{T})$ in the standard setting. In this work, we study the question of whether an algorithm really requires a hint at *every* time step. Somewhat surprisingly, we show that an algorithm can obtain $O(\log T)$ regret with just $O(\sqrt{T})$ hints under a natural query model; in contrast, we also show that $o(\sqrt{T})$ hints cannot guarantee better than $\Omega(\sqrt{T})$ regret. We give two applications of our result, to the well-studied setting of optimistic regret bounds and to the problem of online learning with abstention.

## 1 Introduction

There has been a spate of work on improving the performance of online algorithms with the help of externally available hints. The goal of these works is to circumvent worst-case bounds and exploit the capability of machine-learned models that can potentially provide these hints. There have been two main lines of study. The first is for combinatorial problems, where the goal has been to be improve the competitive ratio of online algorithms; problems considered here include ski-rental [9, 17], caching [13, 19, 26], scheduling [1, 12, 17, 22], matching [16, 18], etc. The second is in the learning theory setting, where the goal has to been to improve the regret of online optimization algorithms. A series of recent papers showed how to achieve better regret guarantees, assuming that we have a hint about the cost function before the algorithm makes a choice. For many variants of the online convex optimization problem, works such as [25, 8, 2, 3] studied the power of having prior information about a cost function. Works such as [29] have also studied improved regret bounds in partial information or bandit settings. In all these works, a desirable property is to ensure *consistency*, which demands better performance with better quality hints, and *robustness*, which guarantees a certain level of performance with poor quality or even adversarially bad hints.

Recall the standard online optimization model [30], which is a game between an algorithm and an adversary. In each round, the algorithm plays a point and the adversary responds with a cost function that is visible to the algorithm, and the cost in this round is measured by evaluating the cost function on the played point. In online linear optimization, the cost function is linear. The regret of an algorithm is the worst-case difference between the cost of the algorithm and the cost of an algorithm

35th Conference on Neural Information Processing Systems (NeurIPS 2021).

that plays a fixed point in each round. A natural way to incorporate hints or prior information is to give the algorithm access to a hint about the cost function at a given round *before* it chooses a point to play at that round. In this sense, hints are present *gratis* [25].

The availability of a hint in each round seems natural in some settings, e.g., when cost functions change gradually with time [25, 4]). However, it can be prohibitive in many others, for instance, if hints are obtained by using expensive side information, or if they are generated by a running a computationally expensive ML model. Furthermore, hints can also be wasteful when the problem instance, or even a large sub-instance, is such that the algorithm cannot really derive any substantial benefit from their presence. This leads to the question of strengthening the online learning with hints model by making it parsimonious. In this work, we pursue this direction, where we offer the algorithm the flexibility to *choose* to ask for a hint before it plays the point. It then becomes an onus on the algorithm to know when to ask for a hint and how to use it judiciously, while ensuring both consistency and robustness. The performance of such an algorithm is measured not only by its regret but also by the number of hints it uses.

**Our contributions.** We now present a high level summary of our results. For ease of exposition, we will defer the formal statements to the respective sections. All our results are for the problem of online *linear* optimization (OLO) when the domain is the unit $\ell_2$ ball; the cost function at every step is defined using a cost vector $c_t$ (and the cost or loss is the inner product of the point played with $c_t$). We call a hint *perfect* if it is the same as the cost vector at that time step, *good* if it is weakly correlated with the cost vector, and *bad* otherwise.

As our main result, we show that for OLO in which hints are guaranteed to be good whenever the algorithm asks for a hint, there is an efficient randomized algorithm that obtains $O(\log T)$ regret using only $O(\sqrt{T})$ hints. We extend our result to the case when $|\mathcal{B}|$ hints can be bad (chosen in an oblivious manner, as we will discuss later), and give an algorithm that achieves a regret bound of $O(\sqrt{|\mathcal{B}|}\log T)$, while still asking for $O(\sqrt{T})$ hints. It is interesting to contrast our result with prior work: Dekel et al. [8] obtained an algorithm with $O(\log T)$ regret, when a good hint is available in every round. Bhaskara et al. [2] made this result robust, obtaining a regret bound of $O(\sqrt{|\mathcal{B}|}\log T)$, when there are $|\mathcal{B}|$ bad hints. Our result improves upon these works by showing the same asymptotic regret bounds, but using only $O(\sqrt{T})$ hints. Our result also has implications for optimistic regret bounds [25] (where we obtain the same results, but with fewer hints) and for online learning with abstention [24] (where we can bound the number of abstentions).

We also show two lower bounds that show the optimality of our algorithm. The first is regarding the minimum number of hints needed to get $O(\log T)$ regret: we show that any (potentially randomized) algorithm that uses $o(\sqrt{T})$ hints will suffer a regret of $\Omega(\sqrt{T})$. The second and more surprising result is the role of randomness: we show that any deterministic algorithm that obtains $O(\log T)$ regret must use $\Omega(T/\log(T))$ hints, even if each of them is perfect. This shows the significance of having a randomized algorithm and an oblivious adversary.

Finally we extend our results to the unconstrained OLO setting (see Section 6), where we design a deterministic algorithm to obtain $O(\log^{3/2} T)$ regret (suitably defined for the unconstrained case) when all hints are good, and a randomized algorithm to obtain $O(\log T)$ regret, which can be extended to the presence of bad hints.

There are three aspects of our results that we find surprising. The first is even the possibility of obtaining $O(\log T)$ regret using only a sublinear number of hints. The second is the sharp threshold on the number of hints needed to obtain logarithmic regret; the regret does not gracefully degrade when the number of hints is below $O(\sqrt{T})$. The third is the deterministic vs randomized separation between the constrained and the unconstrained cases, when all queried hints are good.

We now present some intuition why our result is plausible. Consider the standard "worst-case" adversary for OLO: random mean-zero costs. This case is "hard" because the learner achieves zero expected cost, but the competitor achieves $-\sqrt{T}$ total cost. However, if we simply play a hint $-h_t$ on $O(\sqrt{T}/\alpha)$ rounds, each such round incurs $-\alpha$ cost, which is enough to cancel out the $\sqrt{T}$ regret while making only $O(\sqrt{T})$ hint queries. Thus, the standard worst-case instances are actually *easy* with hints. More details on this example and an outline of our algorithm are provided at the start of Section 3.

**Organization.** Section 2 provides the necessary background. The main algorithm and analysis for the constrained OLO case are in Section 3. The extensions and applications of this can be found in Section 4. Section 5 contains the lower bounds and Section 6 contains the algorithms and analyses for the unconstrained case. All missing proofs are in the Supplementary Material.

## 2 Preliminaries

Let $\|\cdot\|$ denote the $\ell_2$-norm and $\mathbb{B}^d = \{x \in \mathbb{R}^d \mid \|x\| \leq 1\}$ denote the unit $\ell_2$ ball in $\mathbb{R}^d$. We use the compressed sum notation and use $c_{1:t}$ to denote $\sum_{i=1}^{t} c_i$ and $\|c\|_{1:t}^2$ to denote $\sum_{i=1}^{t} \|c_i\|^2$. Let $\vec{c} = c_1, \ldots, c_T$ be a sequence of cost vectors. Let $[T] = \{1, \ldots, T\}$.

**OLO problem and Regret.** The *constrained online linear optimization* (OLO) problem is modeled as a game over $T$ rounds. At each time $t \in [T]$, an algorithm $\mathcal{A}$ plays a vector $x_t \in \mathbb{B}^d$, and then an adversary responds with a *cost vector* $c_t \in \mathbb{B}^d$. The algorithm incurs cost (or loss) $\langle x_t, c_t \rangle$ at time $t$. The total cost incurred by the algorithm is $\mathrm{cost}_{\mathcal{A}}(\vec{c}) = \sum_{t=1}^{T} \langle x_t, c_t \rangle$. The regret of the algorithm $\mathcal{A}$ with respect to a 'comparator' or benchmark vector $u \in \mathbb{B}^d$ is

$$\mathcal{R}_{\mathcal{A}}(u, \vec{c}) = \mathrm{cost}_{\mathcal{A}}(\vec{c}) - \mathrm{cost}_{\mathcal{A}_u}(\vec{c}) = \sum_{t=1}^{T} \langle x_t - u, c_t \rangle,$$

where $\mathcal{A}_u$ is the algorithm that always plays $u$ at every time step. The *regret* of an algorithm $\mathcal{A}$ is its worst-case regret with respect to all $u \in \mathbb{B}^d$:

$$\mathcal{R}_{\mathcal{A}}(\vec{c}) = \sup_{u \in \mathbb{B}^d} \mathcal{R}_{\mathcal{A}}(u, \vec{c}).$$

**Hints and query cost.** Let $\alpha > 0$ be fixed and known. In this paper we consider the OLO setting where, at any round $t$ before choosing $x_t$, an algorithm $\mathcal{A}$ is allowed to obtain a *hint* $h_t \in \mathbb{B}^d$. If $\langle h_t, c_t \rangle \geq \alpha \|c_t\|^2$, we say that the hint is $\alpha$-*good*. If $\mathcal{A}$ opts to obtain a hint at time $t$, then it incurs a query cost of $\alpha \|c_t\|^2$; the query cost is $0$ if no hint was obtained at time $t$. The definition of regret stays the same and we denote it by $\mathcal{R}_{\mathcal{A},\alpha}(\cdot)$. The total *query cost* of $\mathcal{A}$ is given by $\mathcal{Q}_{\mathcal{A},\alpha}(\vec{c}) = \sum_{t=1}^{T} \mathbb{1}_t \cdot \alpha \|c_t\|^2$ where $\mathbb{1}_t$ is an indicator function used to denote whether $\mathcal{A}$ queried for a hint at time $t$. Note that the algorithm does not actually know the query cost for a round until the end of the round. If $\mathcal{A}$ is a randomized algorithm, the notions of expected regret and expected query cost follow naturally. We consider the setting when the adversarial choice of the hint $h_t$ and cost vector $c_t$ at time $t$ is oblivious to whether the algorithm queries for a hint at time $t$ but can depend adaptively on all previous decisions.

More generally, we consider the case that some subset of the hints are "bad" in the sense that $\langle c_t, h_t \rangle < \alpha \|c_t\|^2$; we let $\mathcal{B}$ denote the set of such indices $t$. Although we assume $\alpha$ is known to our algorithms, we do *not* assume any information about $\mathcal{B}$. Further, our algorithm is charged $\alpha \|c_t\|^2$ for querying a hint *even* if the hint was bad.

## 3 Main algorithm

**Intuition and outline.** The high-level intuition behind our algorithm is the following: suppose for a moment that $\alpha = 1/2$ and each hint is $\alpha$-correlated with the corresponding cost. Now suppose the cost vectors $c_1, \ldots, c_T$ are random unit vectors, as in the standard tight example for FTRL. In this case, if an algorithm were to make a hint query for the first $4\sqrt{T}$ steps, set $x_t = -h_t$ in those steps, and play FTRL subsequently, then the cost incurred by the algorithm will be less than $-2\sqrt{T}$ in the first $4\sqrt{T}$ steps, and $0$ (in expectation) subsequently. On the other hand, for random vectors, we have $\|c_{1:T}\| \leq 2\sqrt{T}$ with high probability, and thus the best vector in hindsight achieves a total cost $-\|c_{1:T}\| \geq -2\sqrt{T}$. Thus the algorithm above actually incurs regret $\leq 0$.

It turns out that the key to the above argument is $\|c_{1:T}\|$ being small. In fact, suppose that $c_t$ are unit vectors, and assume that $\|c_{1:T}\| \leq T/4$. Now, suppose the algorithm makes a hint query at $10\sqrt{T}$ random indices, sets $x_t = -h_t$ in those steps, and uses FTRL in the other steps. One can show that the cost incurred by the algorithm is $-5\sqrt{T}$ plus the cost of the FTRL steps. Since the cost in the

FTRL steps is within $\sqrt{T}$ of the cost incurred by the competitor $u$, we can show that the regret is once again $\leq 0$. The missing subtlety here is accounting for the cost of $u$ on the query steps, but using the bound on $\|c_{1:T}\|$, this can be adequately controlled if the queries are done at random.

The above outline suggests that the difficult case is $\|c_{1:T}\|$ being *large*. However, it turns out that prior work of Huang et al. [11] showed that when the domain is the unit ball, FTL achieves logarithmic regret if we have $\|c_{1:t}\| \geq \Omega(t)$ for all $t$.

Our algorithm can be viewed as a combination of the two ideas above. If we could identify the largest round $S$ for which $\|c_{1:S}\|$ is small, then we can perform uniform sampling until round $S$ and use FTL subsequently, and hope that we have logarithmic regret overall (although it is not obvious that the two bounds *compose*). The problem with this is that we do not know the value of $S$. We thus view the problem of picking a sampling probability as a one-dimensional OLO problem in itself, and show that an online gradient descent (OGD) algorithm achieves low regret when compared to all sequences that have the structure of being $\delta$ until a certain time and 0 thereafter (which captures the setting above). The overall algorithm thus picks the querying probability using the OGD procedure, and otherwise resorts to FTRL, as in the outline above.

We now present the details of the overall algorithm (Section 3.3) and the two main subroutines it uses (Sections 3.1 and 3.2).

## 3.1 A sharper analysis of FTRL

In this section we consider the classic adaptive *Follow the Regularized Leader* (FTRL) algorithm $\mathcal{A}_{\text{ftrl}}$ (Algorithm 1) and show a regret bound that is better than the usual one, if the length of the aggregate cost vector grows "rapidly" after a certain initial period.

For convenience, let $\sigma_t = \|c_t\|^2$. Define the regularizer terms as $r_0 = 1$ and for $t \geq 1$, let

$$r_t = \sqrt{1 + \sigma_{1:t}} - \sqrt{1 + \sigma_{1:t-1}}. \tag{1}$$

By definition, we have $r_{0:t} = \sqrt{1 + \sigma_{1:t}}$. Furthermore, we have $r_t < 1$ for all $t$, since $\sigma_t = \|c_t\|^2 \leq 1$. The FTRL algorithm $\mathcal{A}_{\text{ftrl}}$ then plays the points $x_1, x_2, \ldots$, which are defined as

$$x_{t+1} = \operatorname*{argmin}_{\|x\| \leq 1} \left\{ \langle c_{1:t}, x \rangle + \frac{r_{0:t}}{2} \|x\|^2 \right\}. \tag{2}$$

---

| **Algorithm 1** Adaptive FTRL $\mathcal{A}_{\text{ftrl}}$. | **Algorithm 2** OGD with shrinking domain $\mathcal{A}_{\text{ogd}}$. |
|---|---|
| $x_1 \leftarrow 0, r_0 \leftarrow 1$ 
 **for** $t = 1, \ldots, T$ **do** 
 $\quad$ Play point $x_t$ 
 $\quad$ Receive cost $c_t$ 
 $\quad r_{0:t} \leftarrow \sqrt{1 + \|c\|_{1:t}^2}$ 
 $\quad x_{t+1} \leftarrow \operatorname*{argmin}_{\|x\| \leq 1} \left\{ \langle c_{1:t}, x \rangle + \frac{r_{0:t}}{2} \|x\|^2 \right\}$ 
 **end for** | **Require:** Parameter $\lambda$ 
 $\quad p_1 \leftarrow 0, D_1 \leftarrow [0,1]$ 
 $\quad$ **for** $t = 1, \ldots, T$ **do** 
 $\quad\quad$ Play point $p_t$ 
 $\quad\quad$ Receive cost $z_t$ and $\sigma_t \leq 1$ 
 $\quad\quad \{\sigma_t$ will eventually be set to $\|c_t\|^2\}$ 
 $\quad\quad D_t \leftarrow [0, \min(1, \frac{\lambda}{\sqrt{1+\sigma_{1:t}}})]$ 
 $\quad\quad \eta_t \leftarrow \frac{\lambda}{1+\sigma_{1:t}}$ 
 $\quad\quad p_{t+1} \leftarrow \Pi_{D_t}(p_t - \eta_t z_t)$, where $\Pi_{D_t}$ is the 
 $\quad\quad$ projection to the interval 
 $\quad$ **end for** |

---

We will show that $\mathcal{A}_{\text{ftrl}}$ satisfies the following refined regret guarantee:

**Theorem 3.1.** *Consider $\mathcal{A}_{\text{ftrl}}$ on a sequence $c_t$ of cost vectors and let $\alpha \in (0,1)$ be any parameter. Suppose that $S$ is an index such that for all $t > S$, $\|c_{1:t}\| \geq \frac{\alpha}{4}(1 + \sigma_{1:t})$ (recall $\sigma_t = \|c_t\|^2$). Then,*

*1. For all $N \in [T]$ and for any $\|u\| \leq 1$, we have*

$$\sum_{t=1}^{N} \langle c_t, x_t - u \rangle \leq 4.5\sqrt{1 + \sigma_{1:N}}.$$

2. *For the index $S$ defined above, we have the refined regret bound:*

$$\mathcal{R}_{\mathcal{A}_{ftrl}}(\vec{c}) \leq \frac{\sqrt{1 + \sigma_{1:S}}}{2} + \frac{18 + 8\log(1 + \sigma_{1:T})}{\alpha} + \|c_{1:S}\| + \sum_{t=1}^{S} \langle c_t, x_t \rangle. \qquad (3)$$

Part (1) of the theorem follows from the standard analysis of FTRL; we include the proof in Appendix B.1 for completeness. Part (2) is a novel contribution, where we show that if $\|c_{1:t}\|$ grows quickly enough, then the "subsequent" regret is small. It can be viewed as a generalization of a result of Huang et al. [11], who proved such a regret bound for $S = 0$.

## 3.2 Switch-once dynamic regret

In this section we show a regret bound against all time-varying comparators of a certain form. More formally, we consider the one-dimensional OLO problem where the costs are $z_t$ and $\lambda \geq 1$ is a known parameter. We also assume that at time $t$, the algorithm is provided with a parameter $\sigma_t \in [0, 1]$ that will give some extra information about the sequence of comparators of interest. Thus the modified OLO game can be described as follows:

- For $t = 1, 2, \ldots$, the algorithm first plays $p_t \in [0, 1]$, and then $z_t, \sigma_t$ are revealed.
- $z_t$ always satisfies $z_t^2 \leq 4\sigma_t$.
- We wish to minimize the regret with respect to a class of comparator sequences $(q_t)_{t=1}^T$ (defined below), i.e., minimize $\sum_t z_t(p_t - q_t)$ over all sequences in the class.

We remark that for the purposes of this subsection, we can think of $\sigma_t$ as $z_t^2$. (The more general setup is needed when we use this in Algorithm 3.)

**Definition 3.2** (Valid-in-hindsight sequences). *We say that a sequence $(q_t)_{t=1}^T$ is valid in hindsight if there exists an $S \in [T]$ and a $\delta \in [0, 1]$ such that*

1. *$q_t = \delta$ for all $t \leq S$ and $q_t = 0$ for $t > S$.*
2. *At the switching point, we have $\delta^2 \leq \frac{\lambda^2}{1 + \sigma_{1:S}}$.*

We now show that a variant of online gradient descent (OGD) with a *shrinking domain* achieves low regret with respect to all valid-in-hindsight sequences; we call this $\mathcal{A}_{ogd}$ (see Algorithm 2).

**Theorem 3.3.** *Let $\lambda \geq 1$ be a given parameter, and $(z_t)_{t=1}^T$ be any sequence of cost values satisfying $z_t^2 \leq 4\sigma_t$. Let $(q_t)_{t=1}^T$ be a valid-in-hindsight sequence. The points $p_t$ produced by $\mathcal{A}_{ogd}$ then satisfy:*

$$\sum_{t=1}^{T} z_t(p_t - q_t) \leq \lambda \left(1 + 3\log(1 + \sigma_{1:T})\right).$$

## 3.3 Full algorithm

In this section we present the full algorithm that utilizes the above two ingredients.

---
**Algorithm 3** Algorithm with hints $\mathcal{A}_{hints}$.

---
**Require:** Parameter $\alpha$
    Initialize an instance of $\mathcal{A}_{ftrl}$ and an instance of $\mathcal{A}_{ogd}$ with parameter $\lambda = 10/\alpha$
    **for** $t = 1, \ldots, T$ **do**
        Receive $p_t$ from $\mathcal{A}_{ogd}$; Receive $x_t$ from $\mathcal{A}_{ftrl}$
        With probability $p_t$, get a hint $h_t$ and play $\hat{x}_t = -h_t$; otherwise, play $\hat{x}_t = x_t$
        Receive $c_t$
        Send $c_t$ to $\mathcal{A}_{ftrl}$ as $t$th cost; Send $z_t = -\alpha\|c_t\|^2 - \langle c_t, x_t \rangle$ and $\sigma_t = \|c_t\|^2$ to $\mathcal{A}_{ogd}$
    **end for**

---

**Theorem 3.4.** *When $\mathcal{B} = \emptyset$,*

$$\mathbb{E}[\mathcal{R}_{\mathcal{A}_{hints}, \alpha}(\vec{c})] \leq \frac{78 + 38\log(1 + \|c\|_{1:T}^2)}{\alpha} \text{ and } \mathbb{E}[\mathcal{Q}_{\mathcal{A}_{hints}, \alpha}(\vec{c})] \leq 20\sqrt{\|c\|_{1:T}^2}.$$

*Proof.* Let us first bound the expected cost of querying the hints. From the description of $\mathcal{A}_{ogd}$ (Algorithm 2), because of the shrinking domain, we have $p_t \leq |D_{t-1}| \leq \frac{10}{\alpha\sqrt{1+\sigma_{1:t-1}}}$. At time $t$, the expected query cost paid by the algorithm is $p_t \cdot \alpha \|c_t\|^2$. Using the above, we can bound this as

$$\sum_{t=1}^{T} p_t \alpha \|c_t\|^2 \leq \sum_{t=1}^{T} \frac{10\sigma_t}{\sqrt{1+\sigma_{1:t-1}}} \leq \sum_{t=1}^{T} \frac{10\sigma_t}{\sqrt{\sigma_{1:t}}} \leq 20\sqrt{\sigma_{1:T}} = 20\sqrt{\|c\|_{1:T}^2},$$

where the last inequality follows from concavity of the square root function (e.g., [20, Lemma 4]).

Now we proceed to the more challenging task of bounding the expected regret. We start by noting that the expected loss on any round $t$ is simply $\mathbb{E}[\langle c_t, \hat{x}_t \rangle] = -p_t \langle c_t, h_t \rangle + (1-p_t)\langle c_t, x_t \rangle$. Therefore the expected regret for a fixed $u$ is:

$$\sum_{t=1}^{T} \mathbb{E}[\langle c_t, \hat{x}_t - u \rangle] = \sum_{t=1}^{T} p_t \langle c_t, -h_t - x_t \rangle + \langle c_t, x_t - u \rangle \leq \sum_{t=1}^{T} p_t(-\alpha\|c_t\|^2 - \langle c_t, x_t \rangle) + \langle c_t, x_t - u \rangle,$$

where we have used the fact that the hint $h_t$ is $\alpha$-good. The main claim is then the following.

**Lemma 3.5.** *For the choice of $p_t, x_t$ defined in $\mathcal{A}_{hints}$ (Algorithm 3), we have*

$$\sum_{t=1}^{T} p_t(-\alpha\|c_t\|^2 - \langle c_t, x_t \rangle) + \langle c_t, x_t - u \rangle \leq \frac{78 + 38\log(1 + \|c\|_{1:T}^2)}{\alpha} = O\left(\frac{1 + \log(1 + \sigma_{1:T})}{\alpha}\right).$$

Assuming this, the bound on expected regret easily follows, completing the proof. $\square$

We now focus on proving Lemma 3.5.

*Proof of Lemma 3.5.* The key idea is to prove the existence of a valid-in-hindsight sequence $(q_t)_{t=1}^{T}$ (Definition 3.2) such that when $p_t$ on the LHS is replaced with $q_t$, the sum is $O(\log(T)/\alpha)$. The guarantee of Algorithm 2 (i.e., Theorem 3.3) then completes the proof. Specifically, since we set $\lambda = 10/\alpha$ and $z_t = (-\alpha\|c_t\|^2 - \langle c_t, x_t \rangle)$, Theorem 3.3 guarantees that for any valid-in-hindsight sequence $(q_t)_{t=1}^{T}$, we have:

$$\sum_{t=1}^{T} p_t(-\alpha\|c_t\|^2 - \langle c_t, x_t \rangle) + \langle c_t, x_t - u \rangle$$

$$\leq \sum_{t=1}^{T} q_t(-\alpha\|c_t\|^2 - \langle c_t, x_t \rangle) + \langle c_t, x_t - u \rangle + \frac{10}{\alpha}\left(1 + 3\log(1 + \|c\|_{1:T}^2)\right). \tag{4}$$

Let us define $S$ to be the largest index in $[T]$ such that $\|c_{1:S}\| \leq \frac{\alpha}{4}(1 + \sigma_{1:S})$. Firstly, by Theorem 3.1 (part 2), for any such index $S$ we have:

$$\sum_{t=1}^{T} \langle c_t, x_t - u \rangle \leq \frac{\sqrt{1 + \sigma_{1:S}}}{2} + \frac{18 + 8\log(1 + \sigma_{1:T})}{\alpha} + \|c_{1:S}\| + \sum_{t=1}^{S} \langle c_t, x_t \rangle. \tag{5}$$

Let $\Delta := \frac{\sqrt{1+\sigma_{1:S}}}{2} + \|c_{1:S}\| + \sum_{t=1}^{S} \langle c_t, x_t \rangle$ denote the "excess" over the logarithmic term. Note that by Theorem 3.1 (part 1), for a vector $u = \frac{-c_{1:S}}{\|c_{1:S}\|}$, we have $\|c_{1:S}\| + \sum_{t=1}^{S} \langle c_t, x_t \rangle = \sum_{t=1}^{S} \langle c_t, x_t - u \rangle \leq 4.5\sqrt{1 + \sigma_{1:S}}$. Thus, we have $\Delta \leq 5\sqrt{1 + \sigma_{1:S}}$.

Now, note that if $1 + \sigma_{1:S} \leq \frac{100}{\alpha^2}$, we have $5\sqrt{1 + \sigma_{1:S}} \leq 50/\alpha$. Thus, by setting $q_t = 0$ for all $t$ (which is clearly valid-in-hindsight), the proof follows. Further, if $\Delta \leq 1$, then clearly we can again set $q_t = 0$ for all $t$ to complete the proof. Thus, we assume in the remainder of the proof that $1 + \sigma_{1:S} > \frac{100}{\alpha^2}$ and $\Delta > 1$.

Our goal now is to construct a valid-in-hindsight switch-once sequence that has value $q_t = \delta \in [0, 1]$ for $t \leq S$ and $q_t = 0$ for $t > S$ such that we also have :

$$\delta\left(\sum_{t=1}^{S} \alpha\sigma_t + \langle c_t, x_t \rangle\right) \geq 5\sqrt{1 + \sigma_{1:S}}. \tag{6}$$

First, let us understand the term in the parentheses on the LHS above. We bound this using the following claim.

*Claim.* $\alpha \cdot \sigma_{1:S} + \sum_{t \leq S} \langle c_t, x_t \rangle \geq \frac{\alpha}{2}(1 + \sigma_{1:S})$.

*Proof of claim.* Suppose that we have $\sum_{t \leq S} \langle c_t, x_t \rangle < \frac{\alpha}{2}(1 - \sigma_{1:S})$. By definition of $S$, we have

$$\Delta = \frac{\sqrt{1 + \sigma_{1:S}}}{2} + \|c_{1:S}\| + \sum_{t=1}^{S} \langle c_t, x_t \rangle \leq \frac{\sqrt{1 + \sigma_{1:S}}}{2} + \frac{\alpha}{4}(1 + \sigma_{1:S}) + \frac{\alpha}{2}(1 - \sigma_{1:S})$$

$$= \frac{\sqrt{1 + \sigma_{1:S}}}{2} - \frac{\alpha}{4}(1 + \sigma_{1:S}) + \alpha.$$

From our assumption that $\sqrt{1 + \sigma_{1:S}} \geq \frac{10}{\alpha}$, the RHS above is $\leq \alpha$, which in turn is at most 1. Since we assumed $\Delta > 1$, this is a contradiction so the claim holds. $\square$

Thus in order to satisfy (6), we simply choose

$$\delta = \frac{10}{\alpha\sqrt{1 + \sigma_{1:S}}}.$$

By assumption, this lies in $[0, 1]$, and further, for $\lambda = 10/\alpha$, the $(q_t)$ defined above is a valid-in-hindsight sequence. Combining the fact that $\Delta \leq 5\sqrt{1 + \sigma_{1:S}}$ with (6), we have that

$$\sum_{t=1}^{T} q_t(-\alpha\|c_t\|^2 - \langle c_t, x_t \rangle) + \langle c_t, x_t - u \rangle \leq \frac{18 + 8\log(1 + \sigma_{1:T})}{\alpha}.$$

Now appealing to the guarantee of Theorem 3.3 as discussed earlier, we can replace $q_t$ with $p_t$ by suffering an additional logarithmic term on the RHS. Combining all these cases completes the proof of Lemma 3.5, and thus also the proof of Theorem 3.4. $\square$

## 4 Extensions and applications

In the following subsections, we extend the analysis of Algorithm 3 to disparate settings: we consider robustness to uninformative or "bad" hints, the more classical "optimistic" regret setting, and online learning with abstention.

### 4.1 Bad hints

First, we extend Theorem 3.4 to consider the case $\mathcal{B} \neq \emptyset$ by carefully accounting for the regret incurred during rounds where $t \in \mathcal{B}$ and making crucial use of the shrinking domain $D_t$.

**Theorem 4.1.** *For any $\mathcal{B}$,*

$$\mathbb{E}[\mathcal{R}_{\mathcal{A}_{hints}, \alpha}(\vec{c})] \leq \frac{78 + 38\log(1 + \|c\|_{1:T}^2)}{\alpha} + 40\sqrt{\sum_{t \in \mathcal{B}} \|c_t\|^2} + \frac{20}{\alpha}\sqrt{\sum_{t \in \mathcal{B}} \|h_t\|^2}\sqrt{\log(1 + \|c\|_{1:T}^2)}$$

$$= O\left(\frac{\sqrt{|\mathcal{B}|}\log T}{\alpha}\right), \quad \text{and} \quad \mathbb{E}[\mathcal{Q}_{\mathcal{A}_{hints}, \alpha}(\vec{c})] \leq 20\sqrt{\|c\|_{1:T}^2}.$$

### 4.2 Optimistic bounds

Next, we show that our results have implications for *optimistic* regret bounds (e.g., [10, 6, 25, 27, 23]). The standard optimistic regret bound takes the form $O(\sqrt{\sum_{t=1}^{T}\|c_t - h_t\|^2})$. We will show that the same result can be obtained (up to logarithmic factors) even while only looking at $O(\sqrt{T})$ hints. The approach is very simple: if we set $\alpha = \frac{1}{4}$, then a little calculation shows that for $t \in \mathcal{B}$, $\|c_t\|^2 + \|h_t\|^2 = O(\|c_t - h_t\|^2)$, so that Theorem 4.1 directly implies the desired result.

**Theorem 4.2.** *Set $\alpha = \frac{1}{4}$. Then*

$$\mathbb{E}[\mathcal{R}_{\mathcal{A}_{hints},\alpha}(\vec{c})] \leq 312 + 152\log(1 + \|c\|_{1:T}^2) + 80\left(1 + \sqrt{\log(1 + \|c\|_{1:T}^2)}\right)\sqrt{\sum_{t \in \mathcal{B}} \|c_t - h_t\|^2}$$

$$= O\left(\log(T) + \sqrt{\sum_{t=1}^T \|c_t - h_t\|^2 \log(T)}\right), \text{ and } \quad \mathbb{E}[\mathcal{Q}_{\mathcal{A}_{hints},\alpha}(\vec{c})] \leq 20\sqrt{\|c\|_{1:T}^2}.$$

### 4.3   Online learning with abstention

Finally, we apply our algorithm to the problem of online learning *with abstentions*. In this variant of the OLO game, instead of being provided with hints, the learner is allowed to "abstain" on any given round. When the learner abstains, it receives a loss of $-\alpha\|c_t\|^2$ for some known $\alpha$ but pays a query cost of $\alpha\|c_t\|^2$. The regret is again the total loss suffered by the learner minus the total loss suffered by the best fixed adversary, which does not abstain. This setting is very similar to the scenario studied by [24] in the expert setting, but in addition to moving from the simplex to the unit ball, we ask for a more detailed guarantee from the learner: it is not allowed to abstain too often, as measured by the query cost. In this setting, our Algorithm 3 works essentially out-of-the-box: every time the algorithm queries a hint, we instead simply choose to abstain. This procedure then guarantees:

**Theorem 4.3.** *In the online learning with abstention model, the variant of Algorithm 3 that abstains whenever the original algorithm would ask for a hint guarantees expected regret at most:*

$$\frac{78 + 38\log(1 + \|c\|_{1:T}^2)}{\alpha} = O\left(\frac{1 + \log(1 + \sigma_{1:T})}{\alpha}\right).$$

*Further, the expected query cost is at most $20\sqrt{\|c\|_{1:T}^2}$.*

*Proof.* Since we abstain with probability $p_t$ and otherwise play $x_t$, the expected regret is $\sum_{t=1}^T -p_t\alpha\|c_t\|^2 + (1 - p_t)\langle c_t, x_t\rangle - \langle c_t, u\rangle$. Thus the regret bound follows directly from Lemma 3.5. The query cost bound follows the identical argument as Theorem 3.4. $\qquad\square$

## 5   Lower bounds

In this section we first show that the regret bound obtained in Theorem 3.4 is essentially tight. Next, we show that randomness is necessary in our algorithms.

**Theorem 5.1.** *Let $\alpha \in (0,1]$ be any parameter, and suppose $\mathcal{A}$ is an algorithm for OLO with hints that makes $o\left(\frac{\sqrt{T}}{\alpha}\right)$ hint queries. Then there exists a sequence of cost vectors $c_t$ and hints $h_t$ of unit length, such that (a) in any round $t$ where a hint is asked, $\langle c_t, h_t\rangle \geq \alpha\|c_t\|^2$, and (b) the regret of $\mathcal{A}$ on this input sequence is $\Omega(\sqrt{T})$.*

*Proof.* We will construct a distribution over inputs $\{c_t, h_t\}$ and argue that any deterministic algorithm has an expected regret $\Omega(\sqrt{T})$ for inputs drawn from this distribution. By the minmax theorem, we then have a lower bound for any (possibly randomized) algorithm $\mathcal{A}$.

We consider two-dimensional inputs. At each step, $h_t$ is chosen to be a uniformly random vector on the unit circle (in $\mathbb{R}^2$). The cost $c_t$ is set to be $\alpha h_t \pm \sqrt{1 - \alpha^2}\, h_t^\perp$, where $h_t^\perp$ is a unit vector orthogonal to $h_t$, and the sign is chosen uniformly at random. Now for any deterministic algorithm, if $\mathcal{A}$ queries $h_t$ at time $t$, then it can play a point $ah_t + bh_t^\perp$, for scalars $a, b$. In expectation, this has inner product $a\alpha$ with $c_t$, and thus the expected cost incurred by $\mathcal{A}$ in this step is $\geq -\alpha$ (since $|a| \leq 1$). If $\mathcal{A}$ does not query $h_t$, then $c_t$ is simply a random unit vector, and thus the expected cost incurred by $\mathcal{A}$ in this step is 0.

Next, consider the value $\min_{\|u\| \leq 1}\sum_t\langle c_t, u\rangle$, i.e., the best cost in hindsight; this is clearly $-\|\sum_t c_t\|$. By construction, $c_t$ is a uniformly random vector on the unit circle in $\mathbb{R}^2$ (and the

choices are independent for different $t$). Thus we have $\mathbb{E}[\|\sum_t c_t\|] \geq \Omega(\sqrt{T})$. (This follows from properties of sums of independent random unit vectors; see Supplementary Material for a proof.)

Thus, if the algorithm makes $K$ queries, then the expected regret is at least $-K\alpha + \Omega(\sqrt{T})$. This quantity is $\Omega(\sqrt{T})$ as long as $K = o\left(\frac{\sqrt{T}}{\alpha}\right)$, thus completing the proof. $\qquad\square$

We next show that for *deterministic* algorithms, making $O(\sqrt{T})$ hint queries is insufficient for obtaining $o(\sqrt{T})$ regret, even if the hints provided are always 1-good (i.e., hints are perfect).

**Theorem 5.2.** *Let $\mathcal{A}$ be any deterministic algorithm for OLO with hints that makes at most $C\sqrt{T} < T/2$ queries, for some parameter $C > 0$. Then there is a sequence cost vectors $c_t$ and hints $h_t$ of unit length such that (a) $h_t = c_t$ whenever $\mathcal{A}$ makes a hint query, and (b) the regret of $\mathcal{A}$ on this input sequence is at least $\frac{\sqrt{T}}{2(1+C)}$.*

We remark that by setting $C$ appropriately, we can also show that for a deterministic algorithm to achieve logarithmic regret, it needs to make $\Omega\left(\frac{T}{\log T}\right)$ queries.

## 6 Unconstrained setting

---
**Algorithm 4** Algorithm with hints (unconstrained case).

---
**Require:** Parameters $\epsilon, \alpha, K$, $d$-dimensional unconstrained OLO algorithm $\mathcal{A}_{\text{unc}}$, one-dimensional unconstrained OLO algorithm $\mathcal{A}_{\text{unc-1D}}$ guaranteeing (7)
   **for** $t = 1, \ldots, T$ **do**

      {Randomized version} $\mathbb{1}_t \leftarrow 1$ with probability $\min\left(1, \frac{K}{\alpha\sqrt{1+\|c\|_{1:t-1}^2}}\right)$; 0 otherwise.

      {Deterministic version} $\mathbb{1}_t \leftarrow 1$ iff $1 + \sum_{\tau=1}^{t-1} i_\tau \langle c_\tau, h_\tau \rangle \leq K\sqrt{1 + \|c\|_{1:t-1}^2}$.

      Receive $w_t \in \mathbb{R}^d$ from $\mathcal{A}_{\text{unc}}$; Receive $y_t \in \mathbb{R}$ from $\mathcal{A}_{\text{unc-1D}}$.
      If $\mathbb{1}_t = 1$, get hint $h_t$
      Play $x_t = w_t - \mathbb{1}_t h_t y_t$; Receive cost $c_t$.
      Send $c_t$ to $\mathcal{A}_{\text{unc}}$ as $t$th cost; Send $g_t = -\mathbb{1}_t \langle h_t, c_t \rangle \in \mathbb{R}$ to $\mathcal{A}_{\text{unc-1D}}$ as $t$th cost.
   **end for**

---

In this section, we consider *unconstrained* online learning in which the domain is all of $\mathbb{R}^d$. In this setting, it is unreasonable to define the regret using a supremum over all comparison points $u \in \mathbb{R}^d$ as this will invariably lead to infinite regret. Instead, we bound the regret as a function of $u$. For example, when hints are *not* available, standard results provide bounds of the form [7, 15, 28, 21, 5]:

$$\sum_{t=1}^T \langle c_t, x_t - u \rangle \leq \epsilon + A\|u\|\sqrt{\sum_{t=1}^T \|c_t\|^2 \log(\|u\|T/\epsilon + 1)} + B\|u\|\log(\|u\|T/\epsilon + 1), \quad (7)$$

for constants $A$ and $B$ and any user-specified $\epsilon$. Using such algorithms as building blocks, we design an algorithm with $O(\sqrt{T})$ expected query cost and for all comparators $u$, regret is $\tilde{O}(\|u\|/\alpha)$.

The algorithm is somewhat simpler than in the constrained case: we take an ordinary algorithm that does not use hints and subtract the hints from its predictions. Intuitively, each subtraction decreases the regret by $\alpha\|c\|^2$, so we need only $O(\sqrt{T})$ such events. With constraints, this is untenable because subtracting the hint might violate the constraint, but there is no problem in the unconstrained setting. Instead, the primary difficulty is that we need the regret to decrease not by $\alpha\|c\|^2$ but by $\alpha\|u\|\|c\|^2$ for some *unknown* $\|u\|$. This is accomplished by learning a scaling factor $y_t$ that is applied to the hints.

Moreover, in the case that all hints are *guaranteed* to be $\alpha$-good, we devise a *deterministic* algorithm for the unconstrained setting. In light of Theorem 5.2, this establishes a surprising separation between the unconstrained and constrained settings. For the deterministic approach, we directly

measure the total query cost and simply query a hint whenever the cost is less than our desired budget. Note that this strategy fails if we allow bad hints as the adversary could provide a bad hint every time we ask for a hint. The full algorithm is presented in Algorithm 4, with the randomized and deterministic analyses provided by Theorems 6.1 and 6.2.

**Theorem 6.1.** *The randomized version of Algorithm 4 guarantees an expected regret at most:*

$$2\epsilon + \tilde{O}\left(\frac{\|u\|\sqrt{\log(\|u\|T/\epsilon)}\left[K + \frac{\log(\|u\|T/\epsilon)\log\log(T\|u\|/\epsilon)}{K} + \sqrt{\sum_{t\in\mathcal{B}}\|h_t\|^2\log(T)}\right]}{\alpha}\right),$$

*with expected query cost at most $2K\sqrt{\|c\|_{1:T}^2}$.*

**Theorem 6.2.** *If $\mathcal{B} = \emptyset$, then the deterministic version of Algorithm 4 guarantees:*

$$\sum_{t=1}^{T}\langle c_t, x_t - u\rangle \le 2\epsilon + O\left(\frac{\|u\|\sqrt{\log(\|u\|T/\epsilon + 1)}}{\alpha} + \frac{\|u\|\log^{3/2}(\|u\|T/\epsilon)\log\log(\|u\|T/\epsilon)}{K}\right),$$

*with a query cost at most $2K\sqrt{\|c\|_{1:T}^2}$.*

# 7 Conclusions

In this paper, we consider OLO where an algorithm has query access to hints, in both the constrained and unconstrained settings. Surprisingly, we show that it is possible to obtain logarithmic expected regret by querying for hints at only $O(\sqrt{T})$ time steps. Our work also demonstrates an intriguing separation between randomized and deterministic algorithms for constrained online learning. While our algorithms need to know $\alpha$, an open question is to obtain an algorithm that can operate without knowing $\alpha$. Extending our model to the bandit setting is also an interesting research direction.

## Acknowledgments and Disclosure of Funding

Aditya Bhaskara acknowledges the support from NSF awards 2047288 and 2008688.

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
