## Supplementary Material

## A   Missing proofs

**Proposition A.1.** *For any arbitrary non-negative real numbers $a_1, \ldots, a_T$, we have*

$$\sum_{t=1}^{T} \frac{a_t}{1 + a_{1:t}} \leq \log(1 + a_{1:T}).$$

*Proof.* For any $a, b > 0$, we have

$$\frac{a}{b + a} = \int_{x=0}^{a} \frac{1}{b + a} dx \leq \int_{x=0}^{a} \frac{1}{b + x} dx = \log(b + a) - \log(b). \tag{8}$$

The proof now follows from induction. The base case of $t = 1$ follows directly from (8) with $a$ set to $a_1$ and $b$ set to 1. Assuming that the inequality holds for $T - 1$, let us consider the induction step.

$$\sum_{t=1}^{T} \frac{a_t}{1 + a_{1:t}} = \frac{a_T}{1 + a_{1:T}} + \sum_{t=1}^{T-1} \frac{a_t}{1 + a_{1:t}} \leq \frac{a_T}{1 + a_{1:T}} + \log(1 + a_{1:T-1}) \leq \log(1 + a_{1:T}),$$

where the last inequality again follows from (8) with $a$ set to $a_T$ and $b$ set to $1 + a_{1:T-1}$. $\qquad\square$

**Proposition A.2.** *Consider any $c \in \mathbb{R}^d$ and $r \geq 0$ and let $y = argmin_{\|x\| \leq 1} \frac{r}{2}\|x\|^2 + \langle c, x \rangle$. Then, if $\|c\| \geq r$, we have $y = \frac{-c}{\|c\|}$.*

*Proof.* Consider $f(x) = \frac{r}{2}\|x\|^2 + \langle c, x \rangle$. For any $\|x\| \leq 1$, we have the following.

$$f(x) \geq \frac{r}{2}\|x\|^2 - \|c\|\|x\| \geq \min_{\|z\| \leq 1} \left( \frac{r}{2}\|z\|^2 - \|c\|\|z\| \right),$$

since $\|c\| \geq r$, it is an easy exercise to verify that the RHS is minimized at $\|z\| = 1$ and thus

$$f(x) \geq \frac{r}{2} - \|c\|.$$

On the other hand, substituting $y = \frac{-c}{\|c\|}$, we have $f(y) = \frac{r}{2} - \|c\|$ and the proposition follows. $\qquad\square$

**Lemma A.3.** *Let $c_1, \ldots, c_n$ be independent random unit vectors in $\mathbb{R}^d$ (distributed uniformly on the sphere), for some parameters $n, d$, and let $Z = \sum_{t=1}^{n} c_t$ Then we have $\mathbb{E}[\|Z\|] \geq \Omega(\sqrt{n})$.*

*Proof.* First, we note that since $c_t$ are independent, we have

$$\mathbb{E}[\|Z\|^2] = \sum_{t=1}^{n} \|c_t\|^2 = n.$$

We also have

$$\mathbb{E}[(\|Z\|^2)^2] = \mathbb{E}\Big[\Big(\sum_i \|c_i\|^2 + \sum_{i \neq j} \langle c_i, c_j \rangle\Big)^2\Big] \leq n^2 + \sum_{i \neq j} \mathbb{E}[\langle c_i, c_j \rangle^2] \leq 2n^2.$$

Thus by applying the Paley–Zygmund inequality to the random variable $\|Z\|^2$, we have $\Pr[\|Z\|^2 \geq n/4] = \Omega(1)$, and thus $\Pr[\|Z\| \geq \sqrt{n}/2] = \Omega(1)$. Thus the expected value is $\Omega(\sqrt{n})$. $\qquad\square$

# B  A sharper analysis of FTRL

Our goal in this section is to prove Theorem 3.1. As a first step, let us define $\psi_t(x) = \langle c_t, x \rangle + \frac{r_t}{2}\|x\|^2$, (with the understanding that $c_0 = 0$) so that by definition, we have

$$x_{t+1} = \underset{\|x\| \leq 1}{\operatorname{argmin}} \psi_{0:t}(x).$$

**Lemma B.1.** *Let $\psi_t, x_t$ be as defined above. Then for any $m \in [T]$ and any vector $u$ with $\|u\| \leq 1$, we have*

$$\psi_{0:m}(x_{m+1}) + \sum_{t=m+1}^{T} \psi_t(x_{t+1}) \leq \psi_{0:T}(u).$$

When $m = 0$, the lemma is usually referred to as the FTL lemma (see e.g., [14]), and is proved by induction. Our proof follows along the same lines.

*Proof.* From the definition of $x_{T+1}$ (as the minimizer), we have

$$\psi_{0:T}(u) \geq \psi_{0:T}(x_{T+1}).$$

Now, we can clearly write $\psi_{0:T}(x_{T+1}) = \psi_T(x_{T+1}) + \psi_{0:T-1}(x_{T+1})$. Next, observe that from the definition of $x_T$, we have $\psi_{0:T-1}(x_{T+1}) \geq \psi_{0:T-1}(x_T)$. Plugging this above,

$$\psi_{0:T}(u) \geq \psi_T(x_{T+1}) + \psi_{0:T-1}(x_T).$$

Once again, writing $\psi_{0:T-1}(x_T) = \psi_{T-1}(x_T) + \psi_{0:T-2}(x_T)$ and now using the definition of $x_{T-1}$, we obtain

$$\psi_{0:T}(u) \geq \psi_T(x_{T+1}) + \psi_{T-1}(x_T) + \psi_{0:T-2}(x_{T-1}).$$

Using the same reasoning again, and continuing until we reach the subscript $0{:}m$ in the last term of the RHS, we obtain the desired inequality. $\qquad\square$

We are now ready to prove Theorem 3.1.

*Proof of Theorem 3.1.* Let us focus on Part 2 for now (see Lemma B.4 for Part 1). Note that we can rearrange the bound we wish to prove, i.e., (3), as follows. Let $z$ be the unit vector in the direction of $-c_{1:S}$, so that $-\|c_{1:S}\| = \sum_{t=1}^{S} \langle c_t, z \rangle$. Then (3) can be rewritten as

$$\sum_{t=1}^{S} \langle c_t, z - u \rangle + \sum_{t>S} \langle c_t, x_t - u \rangle \leq \frac{\sqrt{1 + \sigma_{1:S}}}{2} + \frac{18 + 8 \log(1 + \sigma_{1:T})}{\alpha}.$$

As a first step, we observe that $\langle c_{1:S}, z \rangle \leq \langle c_{1:S}, x_{S+1} \rangle$; indeed, $\|x_{S+1}\| \leq 1$ by definition. Thus, it will suffice to prove that

$$\sum_{t=1}^{S} \langle c_t, x_{S+1} - u \rangle + \sum_{t>S} \langle c_t, x_t - u \rangle \leq \frac{\sqrt{1 + \sigma_{1:S}}}{2} + \frac{18 + 8 \log(1 + \sigma_{1:T})}{\alpha}. \tag{9}$$

For proving this, we first appeal to Lemma B.1. Instantiating the lemma with $m = S$ and plugging in the definition of $\psi$, we get

$$\langle c_{0:S}, x_{S+1} \rangle + \frac{r_{0:S}}{2} \|x_{S+1}\|^2 + \sum_{t>S} \langle c_t, x_{t+1} \rangle + \frac{r_t}{2} \|x_{t+1}\|^2 \leq \langle c_{0:T}, u \rangle + \frac{r_{0:T}}{2} \|u\|^2.$$

Noting that $c_0 = 0$ and rearranging, we get:

$$\sum_{t=1}^{S} \langle c_t, x_{S+1} - u \rangle + \sum_{t>S} \langle c_t, x_t - u \rangle$$

$$\leq \frac{r_{0:S}}{2} (\|u\|^2 - \|x_{S+1}\|^2) + \sum_{t>S} \left( \frac{r_t}{2} (\|u\|^2 - \|x_{t+1}\|^2) + \langle c_t, x_t - x_{t+1} \rangle \right)$$

$$\leq \frac{r_{0:S}}{2} + \sum_{t>S} \left( \frac{r_t}{2} (\|u\|^2 - \|x_{t+1}\|^2) + \langle c_t, x_t - x_{t+1} \rangle \right).$$

The LHS matches the quantity we wish to bound in (9), and thus let us analyze the RHS quantity, which we denote by $\mathcal{Q}$.

The next observation is that if $t > S$ and $\sqrt{1 + \sigma_{1:t}} \geq \frac{4}{\alpha}$, then the vector $x_{t+1}$ has norm exactly 1. This can be shown as follows. If $t > S$, by the definition of $S$, we have $\|c_{1:t}\| > \frac{\alpha}{4}(1 + \sigma_{1:t})$. Thus, the vector $-c_{1:t}/\sqrt{1 + \sigma_{1:t}}$ has norm $\geq 1$. From the definition of $x_{t+1}$ (see (2)), this means that the global minimizer (without the constraint $\|x\| \leq 1$) of the quadratic form is a point outside the ball, and thus the minimizer of the constrained problem is its projection, which is thus a unit vector. See Proposition A.2 for further details. We next have the following claim.

*Claim.* Let $M$ be the smallest index $> S$ for which $\sqrt{1 + \sigma_{1:M}} \geq \frac{4}{\alpha}$. Then

$$\sqrt{1 + \sigma_{1:M-1}} \leq \max\left\{\sqrt{1 + \sigma_{1:S}}, \frac{4}{\alpha}\right\}.$$

The claim follows by a simple case analysis. If $M = S + 1$, then clearly the LHS is $\sqrt{1 + \sigma_{1:S}}$. Otherwise, from the definition of $M$, we have the desired bound.

Let us get back to bounding the quantity $\mathcal{Q}$ defined above. We split the sum into indices $\leq M - 1$ and $\geq M$. The nice consequence of the observation above is that for all $t \geq M$, as $\|x_{t+1}\| = 1$, we have $\|u\|^2 - \|x_{t+1}\|^2 \leq 0$, thus the term disappears. Also, for $t < M$, we use the simple bound $\frac{r_t}{2}(\|u\|^2 - \|x_{t+1}\|^2) \leq \frac{r_t}{2}$. This gives

$$\mathcal{Q} \leq \frac{r_{0:M-1}}{2} + \sum_{t=S+1}^{T} \langle c_t, x_t - x_{t+1}\rangle.$$

Thus we only need to analyze the summation on the RHS. To bound the summation $\sum_{t=S+1}^{T}\langle c_t, x_t - x_{t+1}\rangle$ consider two cases for $M$ separately: either $M = S + 1$ or $M > S + 1$. If $M = S + 1$, then by Proposition B.3, $\sum_{t=S+1}^{T}\langle c_t, x_t - x_{t+1}\rangle \leq \frac{8}{\alpha}\log(1 + \sigma_{1:T})$. Alternatively, if $M > S + 1$, let us break the summation into terms with $t \leq M - 1$ and terms with $t \geq M$. Proposition B.2 lets us bound the sum of the terms corresponding to $t \leq M - 1$ by $4\sqrt{\sigma_{1:M-1}} < 4r_{0:M-1} \leq \frac{16}{\alpha}$, where the last step is by definition of $M$ and using the fact that $M - 1 > S$. Then Proposition B.3 lets us bound the sum of the terms with $t \geq M$ by $\frac{8}{\alpha}\log(1 + \sigma_{1:T})$. Thus in all cases we have:

$$\mathcal{Q} \leq \frac{r_{0:M-1}}{2} + \frac{16}{\alpha} + \frac{8}{\alpha}\log(1 + \sigma_{1:T}) \leq \frac{\sqrt{1 + \sigma_{1:S}}}{2} + \frac{18}{\alpha} + \frac{8}{\alpha}\log(1 + \sigma_{1:T}),$$

where in the last step we used the claim and bounded the maximum with a sum. $\qquad\square$

## B.1 Auxiliary lemmas

**Proposition B.2.** *For any time step $t \leq T$, the iterates of the FTRL procedure satisfy:*

$$\|x_t - x_{t+1}\| \leq \frac{2\|c_t\|}{\sqrt{1 + \sigma_{1:t-1}}}.$$

*Furthermore, in any time interval $[A, B]$ with $1 \leq A \leq B \leq T$, we have*

$$\sum_{t=A}^{B}\langle c_t, x_t - x_{t+1}\rangle \leq 4\left(\sqrt{\sigma_{1:B}} - \sqrt{\sigma_{1:A-1}}\right).$$

*Proof.* Let us first show the first part. Define $\psi_t(x) = \langle c_t, x\rangle + \frac{r_t}{2}\|x\|^2$. We will invoke [20, Lemma 7], using $\phi_1 = \psi_{0:t-1}$ and $\phi_2 = \psi_{0:t}$. We have that $\phi_1$ is 1-strongly convex with respect to the norm given by $\|x\|_{t-1}^2 = r_{0:t-1}\|x\|^2$ and $\psi_t = \phi_2 - \phi_1$ is convex and $2\|c_t\|$ Lipschitz. Then, since $x_t = \arg\min\phi_1$ and $x_{t+1} = \arg\min\phi_2$, [20, Lemma 7] implies:

$$\|x_t - x_{t+1}\| \leq \frac{2\|c_t\|}{r_{0:t-1}} = \frac{2\|c_t\|}{\sqrt{1 + \sigma_{1:t-1}}}.$$

We can then use this to show the "furthermore" part as follows. For any $t$ in the range, we have

$$\langle c_t, x_t - x_{t+1}\rangle \leq \|c_t\|\|x_t - x_{t+1}\| \leq \frac{2\sigma_t}{\sqrt{1 + \sigma_{1:t-1}}} \leq \frac{2\sigma_t}{\sqrt{\sigma_{1:t}}} \leq 2\int_{\sigma_{1:t-1}}^{\sigma_{1:t}}\frac{dy}{\sqrt{y}},$$

where in the third inequality, we used the fact that $\sigma_t \leq 1$, and in the last inequality, we upper bounded the term via an integral over an interval of length $\sigma_t$. Summing this over $t$ in the interval $[A, B]$ thus gives

$$\sum_{t=A}^{B} \langle c_t, x_t - x_{t+1} \rangle \leq 2 \int_{\sigma_{1:A-1}}^{\sigma_{1:B}} \frac{dy}{\sqrt{y}} = 4 \left( \sqrt{\sigma_{1:B}} - \sqrt{\sigma_{1:A-1}} \right). \qquad \square$$

**Proposition B.3.** *Let $S$ be an index such that for all $t > S$, $\|c_{1:t}\| \geq \frac{\alpha}{4}(1 + c_{1:t})$, and let $t > S$ be an index for which the iterates $x_t$ and $x_{t+1}$ of the FTRL procedure are both unit vectors. Then,*

$$\|x_t - x_{t+1}\| \leq \frac{8\|c_t\|}{\alpha(1 + \sigma_{1:t})}.$$

*Furthermore, let $M > S$ be an index such that $\|x_t\| = 1$ for all $t \geq M$. Then,*

$$\sum_{t=M}^{T} \langle c_t, x_t - x_{t+1} \rangle \leq \frac{8}{\alpha} \log(1 + \sigma_{1:T}).$$

*Proof.* For simplicity, let us denote $g_t = c_{1:t-1}$ and $g_{t+1} = c_{1:t}$. If the iterates of FTRL are unit vectors, we have

$$x_t = -\frac{g_t}{\|g_t\|} \; ; \; x_{t+1} = -\frac{g_{t+1}}{\|g_{t+1}\|}.$$

Thus their difference can be bounded as

$$x_{t+1} - x_t = \left( \frac{g_t}{\|g_t\|} - \frac{g_t}{\|g_{t+1}\|} \right) + \left( \frac{g_t}{\|g_{t+1}\|} - \frac{g_{t+1}}{\|g_{t+1}\|} \right).$$

The second term clearly has norm $\leq \frac{\|c_t\|}{\|g_{t+1}\|}$. Let us bound the first term:

$$\|g_t\| \left| \frac{1}{\|g_t\|} - \frac{1}{\|g_{t+1}\|} \right| = \frac{|\, \|g_{t+1}\| - \|g_t\| \,|}{\|g_{t+1}\|} \leq \frac{\|c_t\|}{\|g_{t+1}\|}.$$

Note that in the last step, we used the triangle inequality. Combining the two, we get

$$\|x_{t+1} - x_t\| \leq \frac{2\,\|c_t\|}{\|c_{1:t}\|} \leq \frac{8\,\|c_t\|}{\alpha(1 + \sigma_{1:t})},$$

as desired. Let us now show the "furthermore" part. From our assumptions about $M$, we can appeal to the first part of the proposition, and as before, we have for any $t \geq M$,

$$\langle c_t, x_t - x_{t+1} \rangle \leq \|c_t\| \|x_t - x_{t+1}\| \leq \frac{8\sigma_t}{\alpha(1 + \sigma_{1:t})} \leq \frac{8}{\alpha} \int_{1+\sigma_{1:t-1}}^{1+\sigma_{1:t}} \frac{dy}{y}.$$

Now, summing this inequality over $t \in [M, T]$ gives us

$$\sum_{t=M}^{T} \langle c_t, x_t - x_{t+1} \rangle \leq \frac{8}{\alpha} \int_{1+\sigma_{1:M-1}}^{1+\sigma_{1:M}} \frac{dy}{y} \leq \frac{8}{\alpha} \log(1 + \sigma_{1:T}). \qquad \square$$

The next lemma is a consequence of the standard FTRL analysis. We include its proof for completeness. This is also Part (1) of Theorem 3.1.

**Lemma B.4.** *For the FTRL algorithm described earlier, for all $N \in [T]$ and for any vector $u$ with $\|u\| \leq 1$, we have*

$$\sum_{t=1}^{N} \langle c_t, x_t - u \rangle \leq 4.5\sqrt{1 + \sigma_{1:N}}.$$

*Proof.* Suppose we use Lemma B.1 with $m = 0$ and $T = N$, then we get:

$$\sum_{t=0}^{N} \psi_t(x_{t+1}) \leq \psi_{0:N}(u).$$

Plugging in the value of $\psi_t$,

$$\sum_{t=1}^{N} \langle c_t, x_t - u \rangle \leq \sum_{t=0}^{N} \frac{r_t}{2} \left( \|u\|^2 - \|x_{t+1}\|^2 \right) + \sum_{t=1}^{N} \langle c_t, x_t - x_{t+1} \rangle.$$

Now, we use the naive bound of $r_{0:N}$ for the first summation on the RHS, and use Proposition B.2 to bound the second summation by $r_{0:N}$. This completes the proof. $\qquad \square$

## C Switch-once dynamic regret

**Theorem 3.3.** *Let $\lambda \geq 1$ be a given parameter, and $(z_t)_{t=1}^{T}$ be any sequence of cost values satisfying $z_t^2 \leq 4\sigma_t$. Let $(q_t)_{t=1}^{T}$ be a valid-in-hindsight sequence. The points $p_t$ produced by $\mathcal{A}_{ogd}$ then satisfy:*

$$\sum_{t=1}^{T} z_t (p_t - q_t) \leq \lambda \left( 1 + 3 \log(1 + \sigma_{1:T}) \right).$$

*Proof.* The proof is analogous to that of OGD (e.g., [30]), but we need fresh ideas specific to our setup. First, observe that since $q$ is a valid-in-hindsight sequence, we have $q_t \in D_t$ for all $t$.

Thus, we have

$$(p_{t+1} - q_t)^2 \leq (p_t - \eta_t z_t - q_t)^2 \qquad \text{(since projection only shrinks distances)}$$
$$= (p_t - q_t)^2 - 2\eta_t z_t (p_t - q_t) + \eta_t^2 z_t^2.$$
$$\implies z_t (p_t - q_t) \leq \frac{(p_t - q_t)^2 - (p_{t+1} - q_t)^2}{2\eta_t} + \frac{\eta_t}{2} z_t^2. \tag{10}$$

We now need to sum (10) over $t$. Note that the second term is easier to bound:

$$\sum_{t=1}^{T} \frac{\eta_t}{2} z_t^2 \leq \frac{\lambda}{2} \sum_{t=1}^{T} \frac{4\sigma_t}{1 + \sigma_{1:t}} \leq 2\lambda \, \log(1 + \sigma_{1:T}), \tag{11}$$

where the last inequality uses Proposition A.1. Suppose $S$ is the time step at which the switch occurs in the sequence $q$, and let $\delta$ be $q_1$ (i.e., the value in the non-zero segment). We split the first term as:

$$\sum_{t=1}^{T} \frac{(p_t - q_t)^2 - (p_{t+1} - q_t)^2}{2\eta_t} = \sum_{t \leq S} \frac{(p_t - \delta)^2 - (p_{t+1} - \delta)^2}{2\eta_t} + \sum_{t > S} \frac{p_t^2 - p_{t+1}^2}{2\eta_t}. \tag{12}$$

Next, by setting $\eta_0 = \lambda$, writing

$$\frac{(p_t - \delta)^2 - (p_{t+1} - \delta)^2}{2\eta_t} = \frac{(p_t - \delta)^2}{2\eta_{t-1}} - \frac{(p_{t+1} - \delta)^2}{2\eta_t} + \frac{(p_t - \delta)^2}{2} \left( \frac{1}{\eta_t} - \frac{1}{\eta_{t-1}} \right),$$

and noting that $\frac{1}{\eta_t} - \frac{1}{\eta_{t-1}} = \frac{\sigma_t}{\lambda}$, we can make the summation telescope. Doing a similar manipulation for the sum over $t > S$, the RHS of (12) simplifies to:

$$\frac{(p_1 - \delta)^2}{2\eta_0} - \frac{(p_{S+1} - \delta)^2}{2\eta_S} + \frac{p_{S+1}^2}{2\eta_S} - \frac{p_{T+1}^2}{2\eta_T} + \sum_{t \leq S} \frac{(p_t - \delta)^2 \sigma_t}{2\lambda} + \sum_{t > S} \frac{p_t^2 \sigma_t}{2\lambda}$$

$$\leq \frac{1}{2\eta_0} + \frac{|D_S|^2}{2\eta_S} + \sum_{t=1}^{T} \frac{|D_t|^2 \sigma_t}{2\lambda}, \tag{13}$$

where $|D_t|$ is the length/diameter of the domain at time $t$, i.e., $|D_t|^2 = \min(1, \frac{\lambda^2}{1 + \sigma_{1:t}})$. The inequality holds because for all $t$, both $p_t$ and $q_t$ are in $D_t$. Plugging in the values of $|D_t|$ and $\eta_t$, the first two terms in (13) are at most $\lambda/2$ (because $\lambda \geq 1$). Thus plugging this back into (12), we get

$$\sum_{t=1}^{T} \frac{(p_t - q_t)^2 - (p_{t+1} - q_t)^2}{2\eta_t} \leq \lambda \left( 1 + \sum_{t=1}^{T} \frac{\sigma_t}{2(1 + \sigma_{1:t})} \right).$$

Finally, using Proposition A.1, the RHS above can be upper bounded by $\lambda \left( 1 + \frac{1}{2} \log(1 + \sigma_{1:T}) \right)$.

Plugging this back into (10), summing over $t$, and using (11), we get

$$\sum_{t} z_t (p_t - q_t) \leq \lambda \left( 1 + 3 \log(1 + \sigma_{1:T}) \right). \qquad \square$$

# D  Proofs for Section 4

**Theorem 4.1.** *For any $\mathcal{B}$,*

$$\mathbb{E}[\mathcal{R}_{\mathcal{A}_{hints},\alpha}(\vec{c})] \leq \frac{78 + 38\log(1 + \|c\|_{1:T}^2)}{\alpha} + 40\sqrt{\sum_{t\in\mathcal{B}}\|c_t\|^2} + \frac{20}{\alpha}\sqrt{\sum_{t\in\mathcal{B}}\|h_t\|^2}\sqrt{\log(1 + \|c\|_{1:T}^2)}$$

$$= O\left(\frac{\sqrt{|\mathcal{B}|}\log T}{\alpha}\right), \quad \text{and} \quad \mathbb{E}[\mathcal{Q}_{\mathcal{A}_{hints},\alpha}(\vec{c})] \leq 20\sqrt{\|c\|_{1:T}^2}.$$

*Proof.* In the proof of Theorem 3.4, we exploited the fact that Lemma 3.5 actually bounds the expected regret when $\mathcal{B} = \emptyset$. However, when $\mathcal{B} \neq \emptyset$, we have a more complicated relationship:

$$\sum_{t=1}^{T}\mathbb{E}[\langle c_t, \hat{x}_t - u\rangle] = \sum_{t=1}^{T} p_t\langle c_t, -h_t - x_t\rangle + \langle c_t, x_t - u\rangle$$

$$\leq \sum_{t\notin\mathcal{B}} p_t(-\alpha\|c_t\|^2 - \langle c_t, x_t\rangle) + \langle c_t, x_t - u\rangle + \sum_{t\in\mathcal{B}} p_t\langle c_t, -h_t - x_t\rangle + \langle c_t, x_t - u\rangle$$

$$= \sum_{t=1}^{T} p_t(-\alpha\|c_t\|^2 - \langle c_t, x_t\rangle) + \langle c_t, x_t - u\rangle + \sum_{t\in\mathcal{B}} -p_t(\langle c_t, h_t\rangle - \alpha\|c_t\|^2)$$

$$\leq \sum_{t=1}^{T} p_t(-\alpha\|c_t\|^2 - \langle c_t, x_t\rangle) + \langle c_t, x_t - u\rangle + \sum_{t\in\mathcal{B}} |D_{t-1}|(\|c_t\|\|h_t\| + \alpha\|c_t\|^2),$$

where $|D_{t-1}| = \frac{10}{\alpha\sqrt{1+\|c\|_{1:t-1}^2}}$, and the last line follows from the restrictions on $p_t$ in Algorithm 2.
The first sum in the above expression is already controlled by Lemma 3.5. For the second sum,

$$\sum_{t\in\mathcal{B}} |D_{t-1}|(\|c_t\|\|h_t\| + \alpha\|c_t\|^2) \leq 2\sum_{t\in\mathcal{B}} |D_t|(\|c_t\|\|h_t\| + \alpha\|c_t\|^2)$$

$$\leq 2\sum_{t\in\mathcal{B}} \frac{10\|c_t\|^2}{\sqrt{1 + \sum_{\tau\in\mathcal{B},\tau\leq t}\|c_\tau\|^2}} + |D_t|\|c_t\|\|h_t\|$$

$$\leq 40\sqrt{\sum_{t\in\mathcal{B}}\|c_t\|^2} + 2\sum_{t\in\mathcal{B}} |D_t|\|c_t\|\|h_t\|$$

$$\text{(by Cauchy–Schwarz)} \leq 40\sqrt{\sum_{t\in\mathcal{B}}\|c_t\|^2} + 2\sqrt{\sum_{t\in\mathcal{B}}\|h_t\|^2}\sqrt{\sum_{t\in\mathcal{B}}\|c_t\|^2|D_t|^2}$$

$$\leq 40\sqrt{\sum_{t\in\mathcal{B}}\|c_t\|^2} + \frac{20}{\alpha}\sqrt{\sum_{t\in\mathcal{B}}\|h_t\|^2}\sqrt{\log(1 + \|c\|_{1:T}^2)}. \qquad \square$$

**Theorem 4.2.** *Set $\alpha = \frac{1}{4}$. Then*

$$\mathbb{E}[\mathcal{R}_{\mathcal{A}_{hints},\alpha}(\vec{c})] \leq 312 + 152\log(1 + \|c\|_{1:T}^2) + 80\left(1 + \sqrt{\log(1 + \|c\|_{1:T}^2)}\right)\sqrt{\sum_{t\in\mathcal{B}}\|c_t - h_t\|^2}$$

$$= O\left(\log(T) + \sqrt{\sum_{t=1}^{T}\|c_t - h_t\|^2\log(T)}\right), \quad \text{and} \quad \mathbb{E}[\mathcal{Q}_{\mathcal{A}_{hints},\alpha}(\vec{c})] \leq 20\sqrt{\|c\|_{1:T}^2}.$$

*Proof.* The idea is to get a bound in terms of $\|c_t - h_t\|^2$. Since $\alpha = \frac{1}{4}$, $t \in \mathcal{B}$ is equivalent to $\langle c_t, h_t\rangle \leq \frac{\|c_t\|^2}{4}$. Thus if $t \in \mathcal{B}$:

$$\|c_t - h_t\|^2 = \|c_t\|^2 - 2\langle c_t, h_t\rangle + \|h_t\|^2 \geq \frac{\|c_t\|^2}{2} + \|h_t\|^2.$$

Therefore, we have:

$$40\sqrt{\sum_{t\in\mathcal{B}}\|c_t\|^2} + 80\sqrt{\sum_{t\in\mathcal{B}}\|h_t\|^2\log(1+\|c\|_{1:T}^2)} \le 80(1+\sqrt{\log(1+\|c\|_{1:T}^2)})\sqrt{\sum_{t\in\mathcal{B}}\|c_t - h_t\|^2}.$$

Now, by Theorem 4.1 we have:

$$\mathbb{E}[\mathcal{R}_{\mathcal{A},\alpha}(\vec{c})] \le \frac{78 + 38\log(1+\|c\|_{1:T}^2)}{\alpha} + 40\sqrt{\sum_{t\in\mathcal{B}}\|c_t\|^2} + \frac{20}{\alpha}\sqrt{\sum_{t\in\mathcal{B}}\|h_t\|^2}\sqrt{\log(1+\|c\|_{1:T}^2)}$$

$$\le \frac{78 + 38\log(1+\|c\|_{1:T}^2)}{\alpha} + 80\left(1 + \sqrt{\log(1+\|c\|_{1:T}^2)}\right)\sqrt{\sum_{t\in\mathcal{B}}\|c_t - h_t\|^2}. \qquad \square$$

# E   Proofs for Section 5

**Theorem 5.2.** *Let $\mathcal{A}$ be any deterministic algorithm for OLO with hints that makes at most $C\sqrt{T} < T/2$ queries, for some parameter $C > 0$. Then there is a sequence cost vectors $c_t$ and hints $h_t$ of unit length such that (a) $h_t = c_t$ whenever $\mathcal{A}$ makes a hint query, and (b) the regret of $\mathcal{A}$ on this input sequence is at least $\frac{\sqrt{T}}{2(1+C)}$.*

*Proof.* The main limitation of a deterministic algorithm $\mathcal{A}$ is that even if it adapts to the costs seen so far, the adversary always knows if $\mathcal{A}$ is going to make a hint query in the next step, and in steps where a query will not be made, the adversary knows which $x_t$ will be played by $\mathcal{A}$.

Using this intuition, we define the following four-dimensional instance. For convenience, let $e_0$ be a unit vector in $\mathbb{R}^4$, and let $S$ be the space orthogonal to $e_0$. The adversary constructs the instance iteratively, doing the following for $t = 1, 2, \ldots$:

1. If the algorithm makes a hint query at time $t$, set $h_t = c_t = e_0$.
2. If the algorithm does not make a hint query, then if $x_t$ is the point that will be played by the algorithm, set $c_t$ to be a unit vector in $S$ that is orthogonal to $x_t$ and to $c_1 + \cdots + c_{t-1}$. (Note that since $S$ is a three-dimensional subspace of $\mathbb{R}^4$, this is always feasible.)

For convenience, define $I_t$ to be the set of indices $\le t$ in which the algorithm has asked for a hint. Then we first observe that for all $t$,

$$\left\|\sum_{j\in[t]\setminus I_t} c_j\right\|^2 = t - |I_t|. \qquad (14)$$

This is easy to see, because $c_t$ is always orthogonal to $e_0$, and thus is also orthogonal to $\sum_{j\in[t-1]\setminus I_{t-1}} c_j$. The equality (14) then follows from the Pythagoras theorem.

Thus, suppose the algorithm makes $K$ queries in total (over the course of the $T$ steps). By assumption $K \le C\sqrt{T} < T/2$. Then we have that

$$\left\|\sum_{j\in[T]} c_j\right\|^2 = K^2 + \left\|\sum_{j\in[T]\setminus I_T} c_j\right\|^2 = K^2 + T - K.$$

Thus the optimal vector in hindsight (say $u$) achieves $\sum_{j\in[T]}\langle c_j, u\rangle = -\sqrt{T - K + K^2}$.

Let us next look at the cost of the algorithm. In every step where it makes a hint query, the best cost that $\mathcal{A}$ can achieve is $-1$ (by playing $-e_0$). In the other steps, the construction ensures that the cost is 0. Thus the regret is at least

$$-K + \sqrt{T - K + K^2} = \frac{T - K}{K + \sqrt{T - K + K^2}} > \frac{T/2}{K + \sqrt{T}} \ge \frac{\sqrt{T}}{2(1+C)}. \qquad \square$$

# F    Proofs for Section 6

In order to prove Theorems 6.1 and 6.2, we first provide the following technical statement that allows us to unify much the analysis:

**Lemma F.1.** *Suppose that $\mathcal{A}_{unc}$ is an unconstrained online linear optimization algorithm that outputs $w_t \in \mathbb{R}^d$ in response to costs $c_1, \ldots, c_{t-1} \in \mathbb{R}^d$ satisfying $\|c_\tau\| \le 1$ for all $\tau$ and guarantees for some constants $A$ and $B$ for all $u \in \mathbb{R}^d$:*

$$\mathcal{R}_{\mathcal{A}_{unc}}(u, \vec{c}) \le \epsilon + A\|u\|\sqrt{\sum_{t=1}^{T} \|c_t\|^2 \log(\|u\|T/\epsilon + 1)} + B\|u\| \log(\|u\|T/\epsilon + 1),$$

*where $\epsilon$ is an arbitrary user-specified constant. Further, suppose $\mathcal{A}_{unc\text{-}1D}$ is an unconstrained online linear optimization algorithm that outputs $y_t \in \mathbb{R}$ in response to $g_1, \ldots, g_{t-1} \in \mathbb{R}$ satisfying $|g_\tau| \le 1$ for all $\tau$ and guarantees for all $y_\star \in \mathbb{R}$:*

$$\sum_{t=1}^{T} g_t(y_t - y_\star) \le \epsilon + A|y_\star|\sqrt{\sum_{t=1}^{T} g_t^2 \log(|y_\star|T/\epsilon + 1)} + B|y_\star| \log(|y_\star|T/\epsilon + 1).$$

*Finally, suppose also that $\mathbb{E}\left[\sum_{t=1}^{T} \mathbb{1}_t|\langle c_t, h_t\rangle|\right] \ge M\sqrt{1 + \|c\|_{1:T}^2} - N$ and $\mathbb{E}\left[\sum_{t \in \mathcal{B}} \mathbb{1}_t|\langle c_t, h_t\rangle|\right] \le H$ and $\mathbb{E}\left[\sum_{t=1}^{T} \mathbb{1}_t\langle c_t, h_t\rangle^2\right] \le F\sqrt{1 + \|c\|_{1:T}^2}$ for some constant $M, N, H, F$. Then both the deterministic and randomized version of Algorithm 4 guarantee:*

$$\mathbb{E}\left[\mathcal{R}_{\mathcal{A}_{unc}}(u, \vec{c})\right] \le 2\epsilon + B\|u\| \log(\|u\|T/\epsilon + 1) + \frac{4A\|u\|(H + N)\sqrt{\log(\|u\|T/\epsilon + 1)}}{M}$$

$$+ \frac{2AB\|u\|\sqrt{\log(\|u\|T/\epsilon + 1)} \log(2A\|u\|T\sqrt{\log(\|u\|T/\epsilon + 1)}/(M\epsilon) + 1)}{M}$$

$$+ \frac{2A^3 F\|u\|\sqrt{\log(\|u\|T/\epsilon + 1)} \log(2A\|u\|T\sqrt{\log(\|u\|T/\epsilon + 1)}/(M\epsilon) + 1)}{M^2}.$$

*Proof of Lemma F.1.* Some algebraic manipulation of the regret definition yields:

$$\mathbb{E}[\mathcal{R}_{\mathcal{A}_{unc}}(u, \vec{c})] \le \mathbb{E}\left[\inf_{y_\star} \sum_{t=1}^{T} \langle c_t, w_t - u\rangle - y_\star \sum_{t=1}^{T} \mathbb{1}_t\langle h_t, c_t\rangle - \sum_{t=1}^{T} \mathbb{1}_t\langle h_t, c_t\rangle(y_t - y_\star)\right]$$

$$\le \mathbb{E}\left[\inf_{y_\star \ge 0} \sum_{t=1}^{T} \langle c_t, w_t - u\rangle - y_\star \sum_{t=1}^{T} \mathbb{1}_t|\langle h_t, c_t\rangle| + 2y_\star \sum_{t \in \mathcal{B}} \mathbb{1}_t|\langle h_t, c_t\rangle| - \sum_{t=1}^{T} \mathbb{1}_t\langle h_t, c_t\rangle(y_t - y_\star)\right].$$

Now using the hypothesized bounds we have

$$\mathbb{E}\left[\mathcal{R}_{\mathcal{A}_{unc}}(u, \vec{c})\right] \le \mathbb{E}\left[\inf_{y_\star \ge 0} \sum_{t=1}^{T} \langle c_t, w_t - u\rangle - y_\star M\sqrt{1 + \|c\|_{1:T}^2} + 2y_\star H + y_\star N - \sum_{t=1}^{T} \mathbb{1}_t\langle h_t, c_t\rangle(y_t - y_\star)\right]$$

$$\le \inf_{y_\star \ge 0} \mathbb{E}\left[2\epsilon + A\|u\|\sqrt{\sum_{t=1}^{T} \|c_t\|^2 \log(\|u\|T/\epsilon + 1)} + B\|u\| \log(\|u\|T/\epsilon + 1)\right.$$

$$\left. - y_\star M\sqrt{1 + \|c\|_{1:T}^2} + 2y_\star H + y_\star N + Ay_\star \sqrt{\sum_{t=1}^{t} g_t^2 \log(y_\star T/\epsilon + 1)} + By_\star \log(y_\star T/\epsilon + 1)\right]$$

using Jensen inequality,

$$\leq \inf_{y_\star \geq 0} 2\epsilon + A\|u\| \sqrt{\sum_{t=1}^{T} \|c_t\|^2 \log(\|u\|T/\epsilon + 1)} + B\|u\| \log(\|u\|T/\epsilon + 1) - y_\star M \sqrt{1 + \|c\|_{1:T}^2}$$

$$+ 2y_\star H + y_\star N + Ay_\star \sqrt{\mathbb{E}\left[\sum_{t=1}^{t} \mathbb{1}_t \langle c_t, h_t \rangle^2\right] \log(y_\star T/\epsilon + 1)} + By_\star \log(y_\star T/\epsilon + 1)$$

$$\leq \inf_{y_\star \geq 0} 2\epsilon + A\|u\| \sqrt{\sum_{t=1}^{T} \|c_t\|^2 \log(\|u\|T/\epsilon + 1)} + B\|u\| \log(\|u\|T/\epsilon + 1) - y_\star M \sqrt{1 + \|c\|_{1:T}^2}$$

$$+ 2y_\star H + y_\star N + Ay_\star \sqrt{F\sqrt{1 + \|c\|_{1:T}^2} \log(y_\star T/\epsilon + 1)} + By_\star \log(y_\star T/\epsilon + 1)$$

with a little rearrangement,

$$\leq \inf_{y_\star \geq 0} 2\epsilon + A\|u\| \sqrt{\sum_{t=1}^{T} \|c_t\|^2 \log(\|u\|T/\epsilon + 1)} + B\|u\| \log(\|u\|T/\epsilon + 1) - \frac{y_\star}{2} M \sqrt{\|c\|_{1:T}^2}$$

$$+ 2y_\star H + y_\star N + By_\star \log(y_\star T/\epsilon + 1)$$

$$+ Ay_\star \sqrt{F\sqrt{1 + \|c\|_{1:T}^2} \log(y_\star T/\epsilon + 1)} - \frac{y_\star}{2} M \sqrt{1 + \|c\|_{1:T}^2}$$

$$\leq \inf_{y_\star \geq 0} 2\epsilon + A\|u\| \sqrt{\sum_{t=1}^{T} \|c_t\|^2 \log(\|u\|T/\epsilon + 1)} + B\|u\| \log(\|u\|T/\epsilon + 1) - \frac{y_\star}{2} M \sqrt{\|c\|_{1:T}^2}$$

$$+ 2y_\star H + y_\star N + By_\star \log(y_\star T/\epsilon + 1) + \sup_X Ay_\star \sqrt{FX \log(y_\star T/\epsilon + 1)} - \frac{y_\star}{2} MX$$

$$\leq \inf_{y_\star \geq 0} 2\epsilon + A\|u\| \sqrt{\sum_{t=1}^{T} \|c_t\|^2 \log(\|u\|T/\epsilon + 1)} + B\|u\| \log(\|u\|T/\epsilon + 1) - \frac{y_\star}{2} M \sqrt{\|c\|_{1:T}^2}$$

$$+ 2y_\star H + y_\star N + By_\star \log(y_\star T/\epsilon + 1) + \frac{y_\star A^2 F \log(y_\star T/\epsilon + 1)}{2M}.$$

Now, we set

$$y_\star = \frac{2A\|u\|\sqrt{\log(\|u\|T/\epsilon + 1)}}{M}.$$

This yields

$$\mathbb{E}[\mathcal{R}_{\mathcal{A}_{\text{unc}}}(u, \vec{c})]$$

$$\leq 2\epsilon + B\|u\| \log(\|u\|T/\epsilon + 1) + 2y_\star H + y_\star N + By_\star \log(y_\star T/\epsilon + 1) + \frac{y_\star A^2 F \log(y_\star T/\epsilon + 1)}{2M}$$

$$\leq 2\epsilon + B\|u\| \log(\|u\|T/\epsilon + 1) + \frac{4A\|u\|(H + N)\sqrt{\log(\|u\|T/\epsilon + 1)}}{M}$$

$$+ \frac{2AB\|u\|\sqrt{\log(\|u\|T/\epsilon + 1)} \log(2A\|u\|T\sqrt{\log(\|u\|T/\epsilon + 1)}/(M\epsilon) + 1)}{M}$$

$$+ \frac{2A^3 F\|u\|\sqrt{\log(\|u\|T/\epsilon + 1)} \log(2A\|u\|T\sqrt{\log(\|u\|T/\epsilon + 1)}/(M\epsilon) + 1)}{M^2}. \qquad \square$$

Now, to prove Theorem 6.1, it suffices to instantiate the Lemma. We restate the Theorem below for convenience:

**Theorem 6.1.** *The randomized version of Algorithm 4 guarantees an expected regret at most:*

$$2\epsilon + \tilde{O}\left(\frac{\|u\|\sqrt{\log(\|u\|T/\epsilon)}\left[K + \frac{\log(\|u\|T/\epsilon)\log\log(T\|u\|/\epsilon)}{K} + \sqrt{\sum_{t\in\mathcal{B}}\|h_t\|^2\log(T)}\right]}{\alpha}\right),$$

*with expected query cost at most $2K\sqrt{\|c\|_{1:T}^2}$.*

*Proof.* Define

$$p_t = \min\left(1, \frac{K}{\alpha\sqrt{1 + \|c\|_{1:t}^2}}\right),$$

so that in the randomized version of Algorithm 4, at round $t$, we ask for a hint with probability $p_{t-1}$. Clearly, the expected query cost is:

$$\mathbb{E}\left[\sum_{t=1}^T \mathbb{1}_t\langle c_t, h_t\rangle\right] = \sum_{t=1}^T \alpha p_{t-1}\|c_t\|^2 \leq K\sum_{t=1}^T \frac{\|c_t\|^2}{\sqrt{\|c\|_{1:t}^2}} \leq 2K\sqrt{\|c\|_{1:T}^2}.$$

Now, to bound the regret we consider two cases. First, if $1 + \|c\|_{1:T}^2 \leq \frac{K^2}{\alpha^2}$, then we have:

$$\mathbb{E}[\mathcal{R}_{\mathcal{A}_{\mathrm{unc}}}(u, \vec{c})] \leq \mathbb{E}\left[\sum_{t=1}^T \langle c_t, w_t - u\rangle - \sum_{t=1}^T \mathbb{1}_t\langle c_t, h_t\rangle y_t\right] \leq \mathbb{E}\left[\sum_{t=1}^T \langle c_t, w_t - u\rangle + \sum_{t=1}^T g_t(y_t - 0)\right]$$

$$\leq 2\epsilon + A\|u\|\sqrt{\sum_{t=1}^T \|c_t\|^2 \log(\|u\|T/\epsilon + 1)} + B\|u\|\log(\|u\|T/\epsilon + 1)$$

$$\leq 2\epsilon + \frac{A\|u\|K\sqrt{\log(\|u\|T/\epsilon + 1)}}{\alpha} + B\|u\|\log(\|u\|T/\epsilon + 1),$$

and so the result follows. Thus, we may assume $1 + \|c\|_{1:T}^2 > \frac{K^2}{\alpha^2}$. In this case, we will calculate values for $M$, $H$, and $F$ to use in tandem with Lemma F.1. First,

$$\mathbb{E}\left[\sum_{t=1}^T \mathbb{1}_t\langle c_t, h_t\rangle^2\right] \leq \sum_{t=1}^T p_{t-1}\|c_t\|^2 \leq \frac{K}{\alpha}\sum_{t=1}^T \frac{\|c_t\|^2}{\sqrt{\|c\|_{1:t}^2}} \leq \frac{2K}{\alpha}\sqrt{1 + \|c\|_{1:T}^2}.$$

So that we may take $F = \frac{2K}{\alpha}$. Next, note that $p_T = \frac{K}{\alpha\sqrt{1+\|c\|_{1:T}^2}}$ by our casework assumption. Therefore:

$$-\alpha p_T\|c\|_{1:T}^2 \leq \alpha - \alpha p_T(1 + \|c\|_{1:T}^2) \leq \alpha - K\sqrt{\|c\|_{1:T}^2},$$

so that we may take $M = K$ and $N = \alpha$. Finally,

$$\sum_{t\in\mathbb{B}} p_t|\langle c_t, h_t\rangle| \leq K\sum_{t\in\mathbb{B}} \frac{\|c_t\|\|h_t\|}{\alpha\sqrt{\|c_t\|_{1:t}^2}} \leq \frac{K}{\alpha}\sqrt{\sum_{t\in\mathbb{B}}\frac{\|c_t\|^2}{\|c_t\|_{1:t}^2}\sum_{t\in\mathbb{B}}\|h_t\|^2} \leq \frac{K}{\alpha}\sqrt{\sum_{t\in\mathbb{B}}\|h_t\|^2\log(1 + \|c\|_{1:T}^2)},$$

so that we may take $H = \frac{K}{\alpha}\sqrt{\sum_{t\in\mathbb{B}}\|h_t\|^2\log(1 + \|c\|_{1:T}^2)}$. Then Lemma F.1 implies

$$\mathbb{E}[\mathcal{R}_{\mathcal{A}_{\text{unc}}}(u, \vec{c})] \leq 2\epsilon + B\|u\| \log(\|u\|T/\epsilon + 1) + \frac{4A\|u\|(H + \alpha)\sqrt{\log(\|u\|T/\epsilon + 1)}}{M}$$

$$+ \frac{2AB\|u\|\sqrt{\log(\|u\|T/\epsilon + 1)} \log(2A\|u\|T\sqrt{\log(\|u\|T/\epsilon + 1)}/(M\epsilon) + 1)}{M}$$

$$+ \frac{2A^3F\|u\|\sqrt{\log(\|u\|T/\epsilon + 1)} \log(2A\|u\|T\sqrt{\log(\|u\|T/\epsilon + 1)}/(M\epsilon) + 1)}{M^2}$$

$$\leq 2\epsilon + B\|u\| \log(\|u\|T/\epsilon + 1) + \frac{4A\|u\|\sqrt{\log(\|u\|T/\epsilon + 1)\sum_{t \in \mathbb{B}} \|h_t\|^2 \log(1 + \|c\|_{1:T}^2)}}{\alpha}$$

$$+ \frac{4A\|u\|\alpha\sqrt{\log(\|u\|T/\epsilon + 1)}}{K}$$

$$+ \frac{2AB\|u\|\sqrt{\log(\|u\|T/\epsilon + 1)} \log(2A\|u\|T\sqrt{\log(\|u\|T/\epsilon + 1)}/(K\epsilon) + 1)}{K}$$

$$+ \frac{2A^3\|u\|\sqrt{\log(\|u\|T/\epsilon + 1)} \log(2A\|u\|T\sqrt{\log(\|u\|T/\epsilon + 1)}/(K\epsilon) + 1)}{K\alpha}.$$

Simplifying the expression yields

$$\mathbb{E}[\mathcal{R}_{\mathcal{A}_{\text{unc}}}(u, \vec{c})]$$

$$\leq 2\epsilon + \tilde{O}\left( \frac{\|u\|(\frac{\log(\|u\|T/\epsilon)^{3/2} \log\log(T\|u\|/\epsilon)}{K} + \sqrt{\log(\|u\|T/\epsilon)\sum_{t \in \mathbb{B}} \|h_t\|^2 \log(1 + \|c\|_{1:T}^2)})}{\alpha} \right). \square$$

### F.1 Deterministic version

Before providing the proof of Theorem 6.2, we need the following auxiliary statement.

**Lemma F.2.** *Suppose $\mathbb{B} = \emptyset$. Then for all $t$, the deterministic version of Algorithm 4 guarantees:*

$$\sqrt{\|c\|_{1:T-1}^2} - K - 1 - \frac{K}{2\alpha} \leq \sum_{t=1}^{T} \mathbb{1}_t \langle c_t, h_t \rangle \leq K\sqrt{1 + \|c\|_{1:T-1}^2}.$$

*Proof.* Define $Z_t = 1 + \sum_{t=1}^{T} \mathbb{1}_t \langle c_t, h_t \rangle$ with $Z_0 = 1$. We will instead prove the equivalent bound:

$$K\sqrt{\|c\|_{1:T-1}^2} - K - \frac{K}{2\alpha} \leq Z_T \leq 1 + K\sqrt{1 + \|c\|_{1:T-1}^2}.$$

The upper bound is immediate from the definition of $Z_T$ and the fact that $\langle c_t, h_t \rangle \leq 1$. For the lower bound, we will prove a slightly different statement that we will later show implies the desired result:

$$\text{for all } t \geq 0, Z_t \geq K\sqrt{1 + \|c\|_{1:t}^2} - K \sum_{t' \leq t | \sqrt{\|c\|_{1:t'}^2} \leq \frac{1}{2\alpha}} \frac{\|c_{t'}\|^2}{2\sqrt{\|c\|_{1:t'}^2}}.$$

We proceed by induction. The base case for $t = 0$ is clear from definition of $Z_t$. Suppose the statement holds for some $t$. Then consider two cases, either $Z_t < K\sqrt{1 + \|c\|_{1:t}^2}$ or not. If $Z_t \geq K\sqrt{1 + \|c\|_{1:t}^2}$, then $Z_{t+1} = Z_t \geq K\sqrt{1 + \|c\|_{1:t}^2} \geq K\sqrt{1 + \|c\|_{1:t+1}^2} - K$ and so the statement holds. Alternatively, suppose $Z_t < K\sqrt{1 + \|c\|_{1:t}^2}$. Then:

$$Z_{t+1} = Z_t + \langle c_{t+1}, h_{t+1} \rangle$$

$$\geq K\sqrt{1 + \|c\|_{1:t}^2} - K - \sum_{t' \leq t | \sqrt{\|c\|_{1:t}^2} \leq \frac{K}{2\alpha}} \frac{\|c_{t'}\|^2}{2\sqrt{\|c\|_{1:t'}^2}} + \alpha\|c_{t+1}\|^2$$

$$\geq K\sqrt{1 + \|c\|_{1:t+1}^2} - \frac{K\|c_{t+1}\|^2}{2\sqrt{1 + \|c\|_{1:t}^2}} - K - \sum_{t' \leq t | \sqrt{\|c\|_{1:t}^2} \leq \frac{K}{2\alpha}} \frac{\|c_{t'}\|^2}{2\sqrt{\|c\|_{1:t'}^2}} + \alpha\|c_{t+1}\|^2$$

$$\geq K\sqrt{1 + \|c\|_{1:t+1}^2} - \frac{K\|c_{t+1}\|^2}{2\sqrt{\|c\|_{1:t+1}^2}} - K - \sum_{t' \leq t | \sqrt{\|c\|_{1:t'}^2} \leq \frac{K}{2\alpha}} \frac{\|c_{t'}\|^2}{2\sqrt{\|c\|_{1:t'}^2}} + \alpha\|c_{t+1}\|^2$$

$$\geq \sqrt{1 + \|c\|_{1:t+1}^2} - K - \sum_{t' \leq t+1 | \sqrt{\|c\|_{1:t'}^2} \leq \frac{K}{2\alpha}} \frac{\|c_{t'}\|^2}{2\sqrt{\|c\|_{1:t'}^2}},$$

so that the induction is complete.

Finally, observe that if $\tau$ is the largest index such that $\sqrt{\|c\|_{1:t}^2} \leq \frac{K}{2\alpha}$, then

$$\sum_{t' \leq t+1 | \sqrt{\|c\|_{1:t'}^2} \leq \frac{K}{2\alpha}} \frac{\|c_{t'}\|^2}{2\sqrt{\|c\|_{1:t'}^2}} \leq \sum_{t'=1}^{\tau} \frac{\|c_{t'}\|^2}{2\sqrt{\|c\|_{1:t'}^2}} \leq \sqrt{\|c\|_{1:\tau}^2} \leq \frac{K}{2\alpha}. \quad \square$$

Now we can prove Theorem 6.2:

**Theorem 6.2.** *If $\mathcal{B} = \emptyset$, then the deterministic version of Algorithm 4 guarantees:*

$$\sum_{t=1}^{T} \langle c_t, x_t - u \rangle \leq 2\epsilon + O\left( \frac{\|u\|\sqrt{\log(\|u\|T/\epsilon + 1)}}{\alpha} + \frac{\|u\| \log^{3/2}(\|u\|T/\epsilon) \log\log(\|u\|T/\epsilon)}{K} \right),$$

*with a query cost at most $2K\sqrt{\|c\|_{1:T}^2}$.*

*Proof.* From Lemma F.2 we have that the query cost is at most $K\sqrt{\|c\|_{1:T}^2}$. To bound the regret, we will appeal to Lemma F.1, which requires finding values for $M, N, H, F$. First, again by Lemma F.2, we have:

$$K\sqrt{1 + \|c\|_{1:T}^2} - 3K - 1 - \frac{K}{2\alpha} \leq K\sqrt{\|c\|_{1:T-1}^2} - K - 1 - \frac{K}{2\alpha} \leq \sum_{t=1}^{T} \mathbb{1}_t \langle c_t, h_t \rangle.$$

So that we may set $M = K$ and $N = 3K + 1 + \frac{K}{2\alpha}$. Next, since $\mathbb{B} = \emptyset$, $H = 0$. Finally, since all hints are $\alpha$-good, we have

$$\sum_{t=1}^{T} \mathbb{1}_t \langle c_t, h_t \rangle^2 \leq \sum_{t=1}^{T} \mathbb{1}_t \langle c_t, h_t \rangle \leq K\sqrt{\|c\|_{1:T}^2},$$

so that we may take $F = K$. Therefore, noticing that the expected regret is the actual regret since the algorithm is deterministic, we have

$$\mathcal{R}_{\mathcal{A}_{\text{unc}}}(u, \vec{c}) \leq 2\epsilon + B\|u\| \log(\|u\|T/\epsilon + 1) + \frac{4A\|u\|(H+N)\sqrt{\log(\|u\|T/\epsilon + 1)}}{M}$$

$$+ \frac{2AB\|u\|\sqrt{\log(\|u\|T/\epsilon + 1)}\log(2A\|u\|T\sqrt{\log(\|u\|T/\epsilon + 1)}/(M\epsilon) + 1)}{M}$$

$$+ \frac{2A^3F\|u\|\sqrt{\log(\|u\|T/\epsilon + 1)}\log(2A\|u\|T\sqrt{\log(\|u\|T/\epsilon + 1)}/(M\epsilon) + 1)}{M^2}$$

$$\leq 2\epsilon + B\|u\|\log(\|u\|T/\epsilon + 1) + 4A\|u\|\left(\frac{4}{\alpha} + \frac{1}{K}\right)\sqrt{\log(\|u\|T/\epsilon + 1)}$$

$$+ \frac{2AB\|u\|\sqrt{\log(\|u\|T/\epsilon + 1)}\log(2A\|u\|T\sqrt{\log(\|u\|T/\epsilon + 1)}/(K\epsilon) + 1)}{K}$$

$$+ \frac{2A^3\|u\|\sqrt{\log(\|u\|T/\epsilon + 1)}\log(2A\|u\|T\sqrt{\log(\|u\|T/\epsilon + 1)}/(K\epsilon) + 1)}{K}$$

$$\leq 2\epsilon + O\left(\frac{\|u\|\sqrt{\log(\|u\|T/\epsilon + 1)}}{\alpha} + \frac{\|u\|\log^{3/2}(\|u\|T/\epsilon)\log\log(\|u\|T/\epsilon)}{K}\right). \quad \square$$