# OpenReview forum: "Logarithmic Regret from Sublinear Hints"
_NeurIPS.cc/2021/Conference — NeurIPS 2021 Poster_

### Official Review · Reviewer_4B2U · 2021-07-06

**Rating:** 7
**Confidence:** 3

**Summary:**

This paper studies how to obtain a good regret bound using only sublinear number of hints for online linear optimization. They show that to obtain $O(\ln T)$ regret, only $O(\sqrt{T})$ number of hints are needed. Moreover, they establish a lower bound showing that $\Omega(\sqrt{T})$ number of hints are necessary to achieve $O(\ln T)$ regret. They also extend their results to various settings such as the presence of bad hints, unconstrained linear optimization, etc.

**Ethical Concerns:**

This work is purely theoretically and has no ethical concerns.

**Limitations And Societal Impact:**

This work is purely theoretically and has no foreseeable societal impact.

**Main Review:**

- Originality: this paper studies a new problem about how many hints are needed to achieve good regret bound. The analysis connecting to switch-once dynamic regret is interesting and novel.
- Quality: the paper is technical sound. Results are complete for the studied setting.
- Clarity: it is a well-written paper and is easy to follow. Intuitions are well explained with concrete examples. The proof sketch provided are also helpful in reproducing the results.
- Significance: the problem studied is quite important in my point of view. In real application the number of hints are often limited and could be expensive to obtain. Understanding the minimum number of (good) hints needed to achieve certain regret bound is thus valuable.

Some questions:
- Can the results be strengthened to high probability bound (on regret and the number of queries)?
- Is it possible to obtain problem dependent lower bound? For example, how many hints are necessary if we want to obtain regret bound of the form $O(||u||\sqrt{\sum_{t=1}^T||c_t||^2_{\star}})$?

**Time Spent Reviewing:**

5 hours

---

> ### Author Response · Authors · 2021-08-10
> **Response**
>
> **High probability bound.**
> Great question!  For the high probability bounds, we suspect that a relatively straightforward application of subgaussian concentration bounds would likely show that the number of queries is $O(\sqrt{T})$ with high probability, but the regret seems much tricker. In particular, naive concentration arguments would likely have a floor of $O(\sqrt{T})$ arising from some variance calculation, and it would take some additional arguments to avoid this.
>
> **Problem-dependent lower bound.**
> Once again, great question!  For the problem-dependent lower bound, this is likely possible by adapting the adversaries for problem dependent lower bounds without hints. Intuitively, one might think that the number of hints required is simply $O(\sqrt{\sum_{t=1}^T \|c_t\|_\star^2})$.  We will add these as further directions to explore.

---

### Official Review · Reviewer_iLUL · 2021-07-14

**Rating:** 7
**Confidence:** 2

**Summary:**

The authors study online linear optimization (OLO) in a setting where the online algorithm has access to hints. They show that $O(\sqrt T)$ hints is sufficient to guarantee $O(\log T)$ regret, and show an explicit algorithm which achieves this bound, along with a matching lower bound on the number of hints required to get $O(\log T)$ regret. They also show that randomization is required for this kind of result in the constrained setting, but surprisingly is not required in the unconstrained setting. They also show some guarantees to robustness against bad hints and various extensions to other online learning settings.

**Ethical Concerns:**

No problem here.

**Limitations And Societal Impact:**

No problem here.

**Main Review:**

The paper is well-written. I have a few questions:

1. Can the authors motivate the model for hints a bit? I am aware it is used in several previous papers, but would still like to understand the motivation behind it a bit better. A more natural model, it seems to me, is that the hint is guaranteed to satisfy $\|h_t - c_t\| < \epsilon$. Instead, the guarantee the authors pick is $(h_t, c_t) \geq \alpha \|c_t\|^2$. Can the authors explain the differences between these models? Is one model strictly stronger or weaker, or more realistic in practice? One weakness of this model for hints is that the $O(\log T)$ regret bound only holds if the hints are not charged to the online algorithm's overall cost - if they were charged, then the algorithm would incur $O(\sqrt T)$ regret (since it uses $O(\sqrt T)$ hints). This seems true for any online algorithm which uses this model for hints.

2. Can the authors explain a bit what they mean by optimistic regret? It seems that they mean regret bounds which depend on the data-dependent quantity $\sum_{t=1}^T \|h_t - c_t\|^2$. I think it would be helpful to add some discussion about this metric (I looked at a few of the references the authors listed but the exact definition still wasn't clear to me).

3. I notice that the authors focus on online linear optimization (OLO), instead of the more general online convex optimization (OCO). A standard technique in OCO (which I'm sure the authors are well aware of!) is simply to reduce OCO to OLO by feeding the gradients of the costs into an OLO algorithm. Can that technqiue be used here? Do these results say anything about the more general convex setting?

Ultimately, it seems to me that the main advance in the paper is that they show that $O(\sqrt T)$ hints are both sufficient and necessary to get $O(\log T)$ regret; previous work assumed $O(T)$ hints (hints in every round). Nailing down this threshold is interesting and merits acceptance in my view.

**Time Spent Reviewing:**

2

---

> ### Author Response · Authors · 2021-08-10
> **Response**
>
> **1. Hint Motivation**
> We feel that our notion of “good” hint via correlation is better than considering only distances. To see this, notice a hint satisfying $|h_t-c_t|<\epsilon$ also satisfies $|h_t|^2 -2\langle h_t, c_t\rangle + |c_t|^2<\epsilon^2$ so that $\langle h_t, c_t\rangle > \frac{1}{2}|c_t|^2 - \frac{\epsilon^2}{2}$. Thus, so long as $|c_t|$ is not too small, we should expect any hint $h_t$ that is close to $c_t$ in distance is also well-correlated. Further, if $|c_t|$ is actually small, then we expect that it cannot influence the regret too much anyway. We in fact formalize this idea in Theorem 4.2, in which we show that our same algorithm with no modifications also obtains a regret guarantee of $O(\sqrt{\sum_{t=1}^T |h_t-c_t|^2})$.
>
> **2. Optimistic regret**
> Yes, by optimistic regret, we mean regret bounds that depend on $\sum_t \|h_t - c_t\|^2$ that are typically obtained using optimistic mirror descent (e.g., [25]). We will clarify this in the revision.
>
> **3. Convex setting**
> In the general convex setting some care needs to be taken in deciding what constitutes a “hint”. A natural route, as you suggest, is to reduce OCO to OLO by taking subgradients, in which case we might think a “hint” should be a vector that is correlated with the next subgradient. In this case all our results would carry over naturally to OCO. However, there is a  bit of a subtlety because the subgradient is itself a function of the point played, which is a function of the hint. Thus one might wish to consider specific loss functions (e.g., online linear classification with logistic loss) so that one can concretely model this interdependency. It is definitely an interesting direction to investigate!

---

> > ### Comment · Reviewer_iLUL · 2021-08-29
> > **Thanks for the response!**
> >
> > Thanks for your detailed reply. I quite like this line of work. After reading the other review's and the author's responses, I feel comfortable with my review and am leaving my score unchanged.

---

### Official Review · Reviewer_6Hvo · 2021-07-16

**Rating:** 6
**Confidence:** 3

**Summary:**

 Prior work has shown that in the online linear optimization problem, if at each round, the learner is provided with a hint on the cost prior to its selection by an adversary, then one can achieve a regret of order $O(\log(T))$. The authors consider the question of how many hints the learner requires in order to achieve such a regret rate. They show that $O(\sqrt{T})$ hints are sufficient by proposing an algorithm that achieves $O(\log(T))$. The proposed algorithm is a combination of Online Gradient Descent (OGD) and Follow The Regularized Leader (FTRL). OGD learns adaptively the probability by which the algorithm decides to query a hint or not, while, FTRL proposes which action to play whenever the algorithm decides not to query a hint. Conversely, the authors also show that with $o(\sqrt{T})$ hints, regret is no less than $\Omega(\sqrt{T})$.

**Ethical Concerns:**

I don't see any ethical issues with this paper since the work is mostly theoretical. Perhaps the authors should consider the potential application of their work and if that may flag any ethical concerns.

**Limitations And Societal Impact:**

I don't see really any potential negative societal impact of this work since it is mostly theoretical. Perhaps the authors could consider looking at potential applications and whether these applications bear any negative impact.

**Main Review:**

Overall, this paper is very well presented and the results look clean. The conveyed message is also interesting and clarifies how often should an algorithm query hints in order to achieve $O(\log(T))$ regret. The results seem to be independent of the dimension $d$. Could authors comment on this? Perhaps the dependency would be more pronounced in a bandit setting. Also, does the shape of the action set play any role?

As a minor remark, in the intuition and outline, could the author clarify in what sense $\Vert c_{1:t} \Vert$ is small or large? this is mentioned multiple times, and $\Vert c_{1:t}\Vert$ si compared to linear order in $t$. It would be very enlightening to the reader if the authors could be more precise here.

**Time Spent Reviewing:**

10 hours

---

> ### Author Response · Authors · 2021-08-10
> **Response**
>
> **Dimension independence**
> Our results are indeed independent of the dimension, in line with recent work in this area that did not restrict the number of hints, as well as the standard bounds for online learning without hints at all. In a bandit setting we do suspect to lose this property as we likely would have to pay some dimension dependence in order to “explore” to some degree. However, how to do this exactly seems a rather difficult problem—in fact, it is unclear if it is even possible to use hints to the same level of effectiveness in the bandit setting!
>
> **Shape of action set**
> The shape of the action set indeed plays a large role in our results. We stated our results for the case of the unit ball in $L_2$ norm, but we expect our results to generalize to any strongly-convex action set, as was done in works such as [2, 9, 12]. This restriction on the curvature of the action set is necessary: without it, even allowing for a hint every round we cannot achieve logarithmic regret (see [9]).
>
> **$\| c_{1:t} \|$ small vs large**
> The precise notions of “large” and “small” are rather technical and data-dependent.  They are reflected in our proofs by the time point at which the “switch-once dynamic regret” algorithm does the switch. Intuitively, however, there is some particular $k\approx \alpha$ whose precise value depends on the problem such that we will consider $\|c_{1:t}\|\ge k t$ to be “large”. This is because if $\|c_{1:t}\|\ge k t$ for all $t$, then it turns out that even without any hints, FTRL already gets regret $O(\log(T)/k)\approx O(\log(T)/\alpha)$. Thus, we need only worry about how to use in hints when $\|c_{1:t}\|\le k t$, which is made possible by the fact that the hints are $\alpha$-good.  We will add this remark in the revision.
>
> **Societal impact**
> Thanks for the suggestion.  We will consider online learning applications to see if there could be any negative impact of our work.

---

> > ### Comment · Reviewer_6Hvo · 2021-08-31
> > **Thanks for your responses!**
> >
> > Thanks for your clarifications. I find this work interesting to say the least and after reading the other reviewers' comments and the authors' responses I feel convinced about my initial score and maintain it for the time being.

---

### Official Review · Reviewer_d9oi · 2021-07-16

**Rating:** 6
**Confidence:** 3

**Summary:**

This paper considers the Online Linear Optimization (OLO) problem with additional hints (predictions) in which at every round $t\in[T]$, the algorithm can decide to receive the hint $h_t$ to help choose an action $x_t$. Upon committing to the action, a loss vector $c_t$ is revealed and the algorithm incurs the cost $\langle c_t,x_t \rangle$. In case the algorithm had received the hint, it also incurs a query cost of $\alpha |c_t|^2$ where $\alpha$ is a pre-specified constant. While prior works had shown that having access to good hints at every round leads to improved logarithmic regret bound (instead of the standard $O(\sqrt{T})$ regret bound of OLO), this work shows that achieving such improved regret bounds is possible with only $O(\sqrt{T})$ hints. Moreover, they show that $o(\sqrt{T})$ hints would not be enough, and the $\Omega (\sqrt{T})$ regret bound can not be avoided in that case. Finally, some extensions of the framework (including the one which allows for bad hints) and the corresponding regret bound for each setting are provided.

**Limitations And Societal Impact:**

The authors have briefly discussed the limitations of their work in Section 7. Also, this theoretical work does not raise any potential ethical or societal concerns in my view.

**Main Review:**

Main strengths of the work:
- The framework of learning-augmented online algorithms is very well motivated and obtaining improved performance bounds which are both consistent (great performance when the predictions are good) and robust (not much worse than the standard worst-case algorithm when the predictions are poor) is of potential interest to a wide audience.
- The assumption that hints have limited availability is very well motivated in the paper via examples.
- The proposed algorithm is relatively intuitive and it is easy to notice its novelties compared to prior works.

Main drawbacks of the paper:
- The paper is not well-written and it is a bit hard to follow for someone with limited exposure to the literature on this problem. It would have been helpful to discuss the prior algorithms and techniques in more detail (for the reader to get more familiar with the literature) and compare and contrast the proposed algorithm with them. In particular, many of the proof details in the paper are not really insightful and they could have been moved to the appendix to make room for a more detailed literature review.
- The choice of the query cost and its importance needs to be discussed in more detail. In particular, the authors need to mention the query cost of previous works and compare theirs with them. Also, it is not clear whether there are any hard (or soft) budget constraints on the query costs. In other words, I am not sure what amount of query cost is considered acceptable.
- The paper lacks any numerical examples. Even some simple synthetic examples would have been particularly helpful to highlight the contrast in the regret performance of the regime with logarithmic regret ($O(\sqrt{T})$ hints) and the one with the standard $O(\sqrt{T})$ regret ($o(\sqrt{T})$ hints).
- While the authors have provided lots of intuitions throughout the paper (particularly in Section 1 and Section 3), these explanations are written from the perspective of someone who is completely familiar with the literature and they are not really helping the reader get more understanding of the techniques.
- The definitions of perfect, good, and bad hints are also not motivated in the paper. It is not clear to me how these different regimes are defined and why the choices make sense in practice. In particular, given that the algorithm requires offline access to the value $\alpha$, I am not sure how $\alpha$ is chosen in different real-world examples.

****************
Post-Rebuttal Update: Thanks to the authors for their detailed responses to the raised questions, I was convinced about the majority of the concerns and I will raise my score to 6.

**Time Spent Reviewing:**

2-3 hours

---

> ### Author Response · Authors · 2021-08-10
> **Response**
>
> **Writing**
> In the revision, we will add more details to the Introduction about prior work on optimistic regret bounds, and introduce some of the terms used in greater detail. This was omitted due to space, but we will add at least a brief description to help guide readers through our contributions and place them in the context of prior literature.  Thanks for the suggestion.
>
> **Query cost choice**
> To our knowledge, prior work on this specific problem [2, 9, 25] did not actually consider query costs and required a hint at *every* time step. Our primary contribution is to demonstrate that we can obtain the best known regret bounds but using *much fewer* hints. In a common setting where the costs $c_t$ are unit vectors, our results show that we can obtain $O(\log T)$ regret using $O(\sqrt{T})$ hints.
>
> **Numerical examples**
> Consider the following concrete sequence of cost vectors: $c_t=(a_t, b_t)$ is a random unit vector in 2d, and the hint $h_t$ is either $(a_t, 0)$ or $(0, b_t)$ with equal probability. Then we have $\mathbb{E}[\langle h_t, c_t\rangle]=\frac{\|c_t\|^2}{2}$. Now, any algorithm that does not employ hints must suffer $\Omega(\sqrt{T})$ regret in expectation since it would have $\mathbb{E}[\langle x_t, c_t\rangle] =0$ for all $t$ no matter how $x_t$ is selected, while the comparison point would have total expected cost $\mathbb{E}[\|\sum_{t=1}^T c_t\] = \Theta(\sqrt{T})$. In contrast, consider the algorithm that queries a hint $h_t$ every $\sqrt{T}/2C$ rounds for some appropriate constant $C$ and plays $x_t = -h_t$ (note: this is a simplified version of our algorithm). This algorithm will achieve $-C\sqrt{T}$ cost on the rounds where the hints were queried, leading to overall $0$ regret in expectation if $C$ is chosen large enough to overcome the constant in the $\Theta(\sqrt{T})$. Our algorithm will pursue a similar strategy, to ensure at most logarithmic regret.  In this example, if $T$ is in the millions, the hint-less algorithm will incur a regret in the thousands, whereas the algorithm with hints will only incur a regret in the tens.
>
> **Explanations**
> In the revision we will elaborate on the techniques to make the exposition clearer.
>
> **Definition, $\alpha$**
> A hint $h_t$ is “good” when the correlation bound $\langle h_t, c_t\rangle \ge \alpha \|c_t\|^2$ holds for some specified $\alpha$. By “perfect” hints, we mean that $h_t=c_t$, so that the algorithm actually knows the next cost ahead of time, and a “bad” hint is simply one that is not “good”. These definitions are identical to previous work in this area, and are motivated by the standard idea in ML to use inner products to measure “similarity” between vectors. (Please see the response to Review 3 for a comparison between similarity- and "distance"-based definitions.)
>
> In our current results, the value of $\alpha$ needs to be chosen via some domain knowledge of the problem that will be solved—the user needs to have some expectation about how good the hints will be. Removing this restriction by designing an algorithm that automatically adapts to unknown $\alpha$, as is possible in the setting where we receive a hint in every time step, is an interesting future direction.  Unfortunately, it seems that prior techniques to achieve such a goal do not easily generalize to the case in which we must pay a query cost for the hints. Intuitively, if $\alpha$ is small, the algorithm must ask for hints at a higher rate than when $\alpha$ is large. Thus our algorithm uses $\alpha$ to "calibrate" the rate of querying hints.

---

### Decision · Program_Chairs · 2021-09-27

**Decision:**

Accept (Poster)

**Comment:**

Reviewers all agree that this paper makes solid contribution to advancing the understanding
of online linear optimization with hints. Please do incorporate all the writing suggestions
from the reviews into the final version.